# Digital insights: Analyzing the reproductive intentions and influencing factors among urban women in China through online platforms

Jing Xiang[1], Xuan Sun[2]*

1 Department of Gynecology, Nankai University Hospital, Nankai University, Tianjin, China, 2 Department of Public Administration, Zhou Enlai School of Government, Nankai University,Tianjin, China

* sunxuan@nankai.edu.cn

## Abstract

### Objective

To explore how fertility-related sentiments expressed by urban Chinese women varied over time and were framed in online discourse between 2011 and 2024.

### Methods

This study employed natural language processing (NLP), TF-IDF-based semantic analysis, and multinomial logistic regression to analyze user-generated content from Douyin, Xiaohongshu, and Kuaishou. Sentiment categories (positive, neutral, negative) were classified, and predictive variables such as education, marital status, and urban tier were modeled to identify structural correlates of fertility attitudes.

### Results

Negative fertility intentions accounted for 71.0% of all discourse, rising sharply from 56.8% in 2021 to 79.8% in 2024. Neutral and positive expressions comprised 28.5% and 0.5%, respectively. Economic constraints—including housing costs (TF-IDF = 2.34)—and environmental concerns (TF-IDF = 2.00) predominated in negative sentiment, while child-rearing costs (TF-IDF = 4.50) were central to neutral positions. Regression analysis revealed marriage was associated with lower odds of negative intention (OR=0.229, 95% CI: 0.194–0.271), while postgraduate education (OR=2.819) and residence in first-tier cities (OR=4.05) were linked to higher odds. Socio-cultural pressures were the most influential predictors of negative sentiment (OR=11.11, 95% CI: 9.07–12.12).

### Conclusion

Fertility intentions among urban Chinese women increasingly represent conscious adaptations to complex structural realities rather than simple expressions of personal

**Data availability statement:** All the original data have been uploaded as supplementary materials.

**Funding:** This study was financially supported by the Chey Institute's ISEF program in the form of an award received by XS. This study was also financially supported by the Youth Project of the Humanities and Social Sciences Fund of the Ministry of Education in the form of a grant (19YJC840030) received by JX. This study was also financially supported by the ESRC Global Challenges Research Fund in the form of a grant (IES/P011020/1) received by JX and XS. This study was also financially supported by the National Natural Science Foundation of China in the form of a grant (72074127) received by XS. The funders had no role in study design, data collection and analysis, decision to publish, or preparation of the manuscript.

**Competing interests:** The authors have declared that no competing interests exist.

reluctance. This shift reflects a broader societal transition in which autonomy, well-being, and ethical responsibility gradually supersede traditional reproductive imperatives. Rather than indicating demographic crisis, these changing intentions mark a natural phase of social development, underscoring the need for institutional reforms and a cultural ethos that affirms reproductive freedom and supports parenthood as an informed, empowered choice.

## 1. Background

Global fertility rates have declined markedly in recent decades, prompting growing concern among researchers and policymakers across demographic, economic, and public health domains [1,2]. This trend is particularly acute in East Asia, where countries such as China, Japan, and South Korea face unprecedented demographic challenges related to ultra-low fertility [3–5]. In China, the relaxation of the one-child policy in 2016 failed to reverse the persistent fertility decline. The total fertility rate (TFR) fell to 1.3 in 2020 [6], and despite policy adjustments, it continued to decline, with the United Nations estimating a TFR of 1.00 in 2023 [7]. In the same year, China's birth population fell to 9.02 million, with a crude birth rate of 6.39 per thousand [8]—among the lowest in recorded history—well below the replacement threshold of 2.1 [9]. This persistent decline signals an urgent need to investigate the socio-economic factors influencing reproductive decisions, particularly among young women, which are especially crucial for forecasting broader demographic patterns.

Emerging research suggests that social media plays a growing role in influencing reproductive attitudes, especially among younger women in urban contexts [10,11]. As interactive platforms increasingly mediate information exposure and value formation, they also become key arenas where fertility-related narratives and sentiments are constructed, contested, and shared in real time. In China, five dominant social platforms—Douyin, Kuaishou, Xiaohongshu, Bilibili, and Weibo—reached a collective 1.071 billion monthly active users (MAUs) by the end of 2024 [12]. Douyin's dominance as China's largest short-video platform (743 million Monthly Active Users, or MAU—a metric referring to the number of unique users who engage with a platform at least once within a 30-day period) is undergirded by a 69.2% female user majority [13,14], with 50% of active contributors originating from Tier-1/2 cities. Kuaishou, ranks as the second-largest video application in China, with MAU exceeding 457 million [15], female users represent 44.8% of the platform's total users, with a significant portion falling within the 18–34 age range [16]. Similarly, Xiaohongshu, a leading platform for lifestyle and consumer content, has seen rapid growth, with 199 million MAU in 2023, 68% of whom are women, predominantly within the same age range as Kuaishou [17]. The significant reach of these platforms across diverse demographics presents a unique opportunity to examine the current broader fertility attitudes among china urban women, especially within the context of China's rapidly changing demographic landscape. The exclusion of Weibo (485 million MAU,58%

entertainment-focused ephemeral content) and Bilibili (363 million MAU, 62% male users) is methodologically necessitated by their demonstrable misalignment with sustained fertility discourse [7].

User-generated content (UGC) on social media platforms offers a substantial repository of spontaneous expressions, reflecting individual beliefs, emotional attitudes, and perceived social norms [18]. Prior research has demonstrated that online discourse influences reproductive decision-making both through direct norm-setting and through more diffuse mechanisms of social reinforcement [19,20]. Despite this relevance, digital fertility discourse remains insufficiently explored in population studies, particularly within the Chinese context. To address this gap, the present study draws on the cognitive-social model of fertility intentions [21] to examine large-scale user-generated content from Douyin, Kuaishou, and Xiaohongshu. These platforms serve as discursive arenas where urban Chinese women articulate a wide spectrum of reproductive attitudes grounded in everyday experience. The analysis provides a cross-sectional snapshot of fertility sentiment as expressed by urban Chinese women in digital narratives, offering insight into how their reproductive perspectives are voiced, contextualized, and situated within contemporary social life.

## 2. Methodology

### 2.1. Data filtering

User-generated content (UGC) related to fertility was retrospectively analyzed based on data collected between May 2021 and April 2024, coinciding with the implementation of China's three-child policy and providing a relevant policy backdrop. Formal data extraction commenced in May 2024. Content was retrieved via official APIs using a keyword-based filtering strategy informed by predefined thematic categories. The initial keyword list—including terms such as *"fertility intention," "pregnancy," "family formation,"* and *"work-life balance"*—was iteratively refined to incorporate emerging topics like *"policy impact"* and *"working mothers."* This process yielded content from 23,154 unique user accounts.

As shown in Fig 1, the analytic workflow followed a multi-stage process. Textual materials were directly processed, while video content was transcribed exclusively to extract thematically relevant information. Only publicly spoken content in Mandarin or widely understood dialects was included; personal introductions, identifiable anecdotes, and background conversations were explicitly excluded. Transcribed texts were immediately anonymized by removing any potentially sensitive or identifying language before thematic analysis.

All data were fully de-identified from the outset. No personally identifiable information (PII), including usernames, profile images, or IP addresses, was retained. Geographic data, where available, were aggregated to the municipal or provincial level, and demographic variables were derived solely from publicly disclosed, non-sensitive statements, in strict adherence to platform policies and internationally recognized research ethics standards. The final anonymized dataset is provided as Supporting Information (S1 Appendix).

No attempts were made to reconstruct individual behavioral profiles or infer latent psychological characteristics beyond the scope of publicly available content. This study fully complies with the Personal Information Protection Law (PIPL) of the People's Republic of China and relevant platform privacy policies. Given the exclusive use of anonymized, aggregated public data for population-level analysis, the study presents no foreseeable ethical risks related to individual privacy or data security.

**2.1.1. Gender-based sample selection.** This study focuses on fertility-related discourse produced by female users, who uniquely experience the physiological, psychological, and socioeconomic burdens of childbearing. These lived experiences inform complex reproductive decisions, making women's perspectives particularly relevant to fertility research [22]. Despite advancements in gender equality, women remain primary decision-makers in household fertility planning across diverse cultural contexts [23]. In digital spaces, female users also exhibit higher engagement in fertility discussions, while male participation remains limited, reflecting societal perceptions of reproduction as predominantly a female concern [24,25]. These patterns justify focusing the analysis on female-generated content to more accurately capture the complexities of fertility intentions.

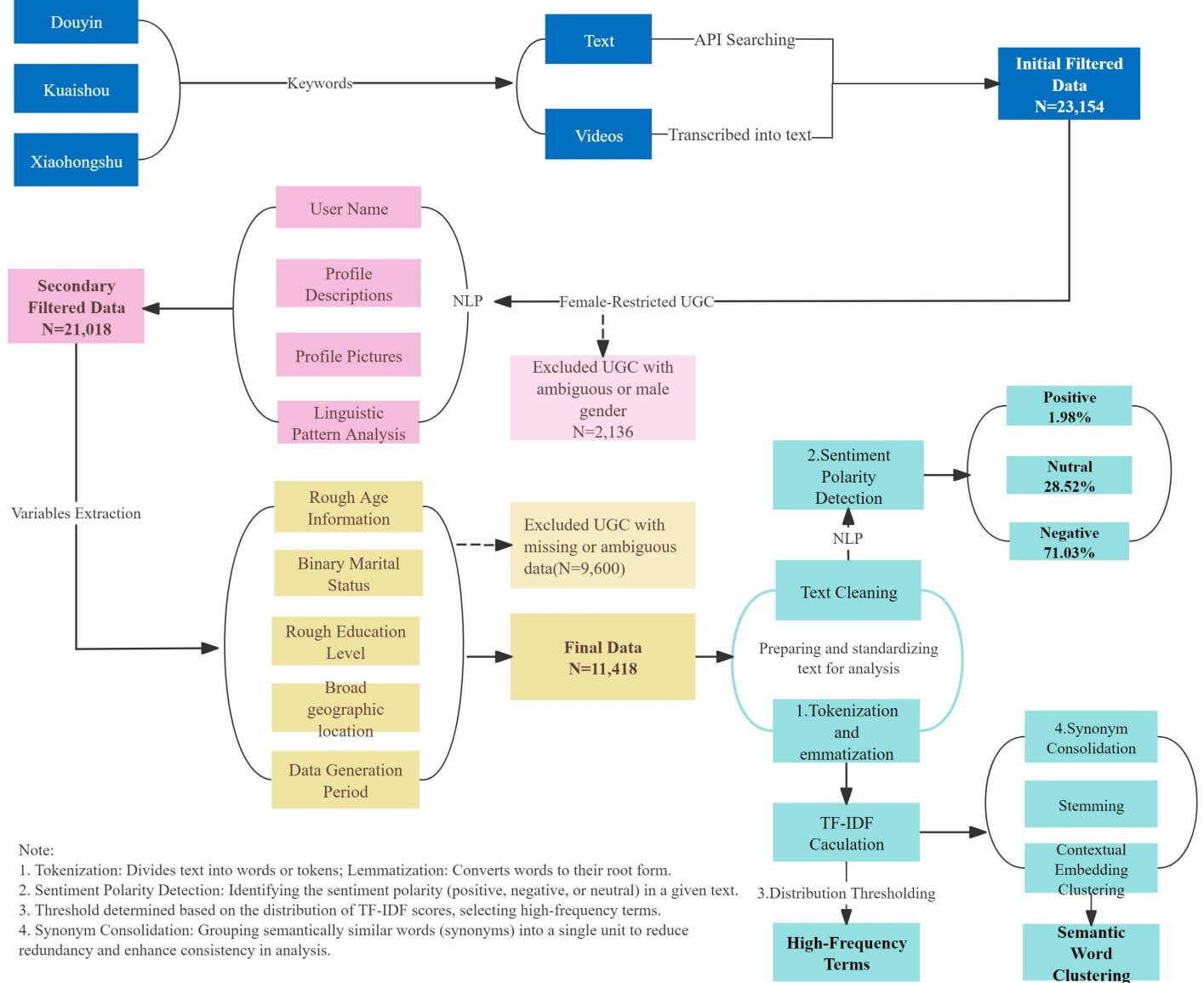

**Fig 1. Multi-stage analytic workflow. Note**:This workflow comprised initial keyword-based data screening, content transcription, semantic preprocessing, TF-IDF term extraction (threshold ≥1.5), and sentiment classification using IBM Watson NLU.

Gender classification employed a multi-step inference protocol. User gender was first inferred from publicly available profile metadata, including usernames, self-descriptions, and avatars, and further validated through NLP-based analysis of linguistic cues and behavioral patterns. In the second-stage filtering, 2,136 users (9.2% of the initial 23,154-user corpus) were excluded due to unreliable gender identification or classification as male or gender-neutral, resulting in a dataset composed exclusively of female users for analysis. At no point were profile images, usernames, or IP addresses stored or linked to analytical datasets during this process.

**2.1.2. Variables extraction.** Age, education, marital status, and region were selected as key explanatory variables based on their consistently documented relevance to fertility intentions in recent demographic research [26–28]. These variables enable stratified analysis of fertility attitudes across socio-demographic and geographic dimensions.

Data were extracted from user-generated content (UGC) using a combination of direct identification or indirect inference via semantic analysis of publicly available posts. All data processing procedures fully complied with the Personal Information Protection Law (PIPL) of the People's Republic of China and the privacy policies of the respective platforms. Personally identifiable information (PII) was neither collected nor retained at any stage. Variable extraction was conducted after complete anonymization and content-level aggregation, ensuring both ethical compliance and analytical reliability.

*Age:* Age was inferred from self-reported temporal cues (e.g., "graduated in 2023") or generational terms (e.g., "95后") within publicly visible posts. These references were mapped to analytical age brackets, such as categorizing "95后" as 26–30 years old within the 20–30 group. This indirect, language-based inference avoids sensitive identifiers and is legally permissible under Article 13 of the PIPL when used for academic research on publicly disclosed data.

*Marital Status:* Relationship status was derived from explicit relational language (e.g., "my husband," "boyfriend"). For statistical modeling, this was binarized into married and single categories, ensuring analytical simplicity while acknowledging relationship diversity. Only contextually grounded, user-disclosed information was considered, avoiding any identity verification or private data linkage.

*Education Level:* Educational attainment was assessed via self-reported status or inferred from occupation-related descriptions. Phrases such as "pursuing a master's degree" or "working on an assembly line" were mapped to three ordinal categories: below bachelor's, bachelor's, and post-bachelor's. All inferences were drawn from public statements and generalized to avoid individual profiling.

*Residence Location:* Geographic location was triangulated using self-declared city names and platform-assigned, de-identified IP metadata. Discrepancies exceeding 50 km between the two sources triggered exclusion (n = 326, 1.4%). IP-derived data were used solely for immediate province-level verification and discarded without storage. Aggregation ensured that no regional identifiers could be traced back to individual users.

*Data Collection Year:* Each UGC entry was timestamped at the time of capture. This metadata enabled the temporal mapping of shifts in discourse between 2021 and 2024 and is legally considered non-sensitive information.

During third-stage filtering, data from 9,600 users were excluded due to incomplete, ambiguous, or contextually unresolvable socio-demographic information. This ensured that the final analytical dataset contained only interpretable, non-identifiable demographic indicators suitable for robust statistical modeling. While inference based on indirect cues introduces some uncertainty, stringent safeguards against re-identification were implemented throughout. All findings should therefore be interpreted within these defined methodological and ethical constraints.

## 2.2. Sample weighting adjustment

To correct for potential biases in platform representation and ensure a more accurate sample, a weighting adjustment based on the Monthly Active Users (MAU) ratio was applied. According to QuestMobile 2024 data, the distribution of users across the three platforms was as follows: Douyin 51.5%, Kuaishou 31.7%, and Xiaohongshu 13.8%. Since the original sample was collected using a non-weighted approach, with Douyin having a disproportionately higher representation, the sample was adjusted based on the relative MAU proportions of each platform.

The adjustment process involved calculating the weight for each platform, ensuring that each platform's representation in the final dataset was proportional to the actual platform user base. The weights were calculated as follows:

$$W_{platform} = \frac{MAU_{platform}}{\sum_{i=1}^{3} MAU_{platform}}$$

Where $W_{platform}$ is the weight for each platform, and $MAU_{platform}$ represents the monthly active users of each platform. This MAU-proportional weighting approach ensures representativeness relative to each platform's population penetration rate.

## 2.3. Definition and analysis

**2.3.1. Theoretical model of fertility intentions.** In this study, fertility intentions are defined as "individuals' expressed plans or desires to have children [29], shaped by both personal attitudes and broader structural influences". This conceptualization integrates the cognitive–social model and the pathway framework [19,30], emphasizing the contextual fluidity of fertility intentions rather than viewing them as fixed behavioral outcomes. Building on this framework, the study analyzes social media discourse to explore how these intentions manifest in real-world narratives. As most users do not explicitly disclose parental status, parity identification is often infeasible unless voluntarily stated. Available data show that 2,154 users (18.9% of the sample) identified as having children, while the majority remain unspecified, resulting in a heterogeneous sample of mothers and non-mothers. Although this limits conventional demographic stratification, it aligns with the study's attitudinal focus on how reproductive perspectives are shaped within broader life narratives, irrespective of actual childbearing status.

These attitudinal expressions provide meaningful insights when interpreted through the Theory of Planned Behavior (TPB) [31], which conceptualizes intention as a proximal predictor of behavior under favorable conditions. Together, these models offer a layered analytical framework: the cognitive–social model addresses attitude formation, the pathway framework explains attitudinal progression, and TPB clarifies the transition from intention to behavior. This integrated approach broadens the understanding of fertility intentions beyond traditional demographic categorizations.

**2.3.2. Sentiment and TF-IDF analysis.** The sentiment analysis framework employed in this study prioritized rigorous text preprocessing to mitigate noise and enhance the semantic fidelity of user-generated content (UGC) related to fertility attitudes. This preparatory phase was foundational to ensuring analytical validity and thematic coherence in downstream sentiment classification.

I. Text Preprocessing

HanLP was employed to segment raw Chinese UGC into linguistically meaningful units, providing the basis for semantic parsing. Stopwords were removed using the HIT Chinese Stopwords list, expanded with fertility-specific terms to suppress high-frequency functional words and platform-specific noise. Informal expressions, including slang and abbreviations, were normalized through a custom mapping system to preserve semantic integrity. Finally, contextual normalization using Baidu's ERNIE embeddings aligned semantically equivalent phrases (e.g., "生育计划" and "组建家庭") into unified vector representations, reducing lexical variation without loss of meaning.

II. Sentiment Classification

Sentiment classification was conducted using IBM Watson's Natural Language Understanding (NLU) platform [32,33], which supports Chinese-language processing through lexical valence dictionaries, negation-sensitive parsing, and contextual embeddings. The system classified text into positive, neutral, and negative categories, accounting for polarity reversals (e.g., negation) and capturing both explicit and nuanced sentiment.

Validation involved benchmarking against the Chinese Sentiment Knowledge Base (CSKB) [34,35], where the model achieved 84.6% accuracy and an F1-score of 0.82 on a test set of 2,000 manually annotated posts. For 12% of the corpus with low-confidence results, sentiment was adjudicated by expert linguistic reviewers, yielding an inter-rater reliability of $\kappa = 0.78$. This hybrid validation minimized classification errors from sarcasm, ambiguity, or figurative speech.

The rigor of this process ensured interpretive clarity for downstream analysis of affective attitudes toward fertility within the UGC corpus.

III. TF-IDF

For thematic analysis of external fertility factors, we conducted automated TF-IDF (Term Frequency-Inverse Document Frequency) [36] analysis through NVivo's algorithmic implementation. This computational method quantifies term significance using a dual weighting mechanism:

i. Term Frequency (TF): Normalized count of a term within individual documents. It is typically calculated as:

$$TF_{(t,d)} = \frac{\text{Number of times term t appears in document d}}{\text{Total number of terms in document d}}$$

ii. Inverse Document Frequency (IDF): A logarithmic penalty is applied to terms appearing across multiple documents:

$$IDF_{(t)} = log(\frac{N}{DF_{(t)} + 1})$$

Where N is the total number of documents in the collection, DF(t) is the number of documents that contain the term t. Laplace smoothing is used to adjust for potential bias caused by term frequency imbalances, with the denominator term adjusted to DF(t)+1 to prevent zero-division errors and reduce statistical instability from out-of-vocabulary and low-frequency terms, such as emerging internet slang in social media data. This approach aligns with standard practices in open-domain text analysis [37,38].

iii. TF-IDF Formula:

$$TF - IDF(t, d) = TF(t, d) \times IDF(t)$$

Table 1 illustrates the application of TF-IDF to fertility-related textual data, emphasizing domain-specific vocabulary and suppressing ubiquitous yet uninformative terms. This generates sparse feature vectors, capturing thematic nuances for further computational analysis, with a dynamically determined TF-IDF threshold of 1.5, derived from score distribution analysis (Fig 2). The dynamic threshold optimizes term retention while minimizing noise, ensuring robust analysis by balancing dimensionality reduction with semantic fidelity, and translating unstructured text into interpretable structured representations.

Semantic clustering was applied to reduce lexical redundancy. Terms were embedded using 300-dimensional fastText vectors and aligned using Chinese WordNet with a cosine similarity threshold ≥ 0.85, consolidating related expressions (e.g., "养育成本" and "经济压力") under broader thematic constructs such as Economic Constraints. All thematic labels were finalized through expert validation.

A three-tiered validation ensured methodological rigor: internal stability testing (92.3% term overlap; Jaccard index = 0.85), algorithm–expert consensus assessment (Cohen's κ = 0.71; 89% agreement), and additional diagnostics detailed in S2 Appendix.

**Table 1. Full-process TF-IDF calculation based on preprocessed fertility discourse.**

| Dataset | Extracted Terms | TF | IDF | TF-IDF Calculation |
|---|---|---|---|---|
| 1."我想要以后有孩子。(I want to have children in the future.)" | want, future, children | 0.33,0.33,0.33 | children = $log(\frac{3}{2+1})$ = 0 want/future = $log(\frac{3}{1+1})$= 0.1761 | children=0.33*0=0 want/future=0.33 *0.1761=0.0581 |
| 2."我觉得抚养孩子太贵了。(I'm afraid that raising children is too expensive.)" | afraid, raise, children, expensive | 0.25,0.25,0.25,0.25 | afraid...expensive = $log(\frac{3}{1+1})$= 0.1761 | afraid...expen-sive=0.25*0.1761=0.044 |
| 3."有家庭很重要，但我还没准备好。(It's important to have a family, but I'm not ready yet.)" | family, important, Not yet, ready | 0.25,0.25,0.25,0.25 | family...ready = $log(\frac{3}{1+1})$= 0.1761 | family... ready=0.25*0.1761=0.044 |

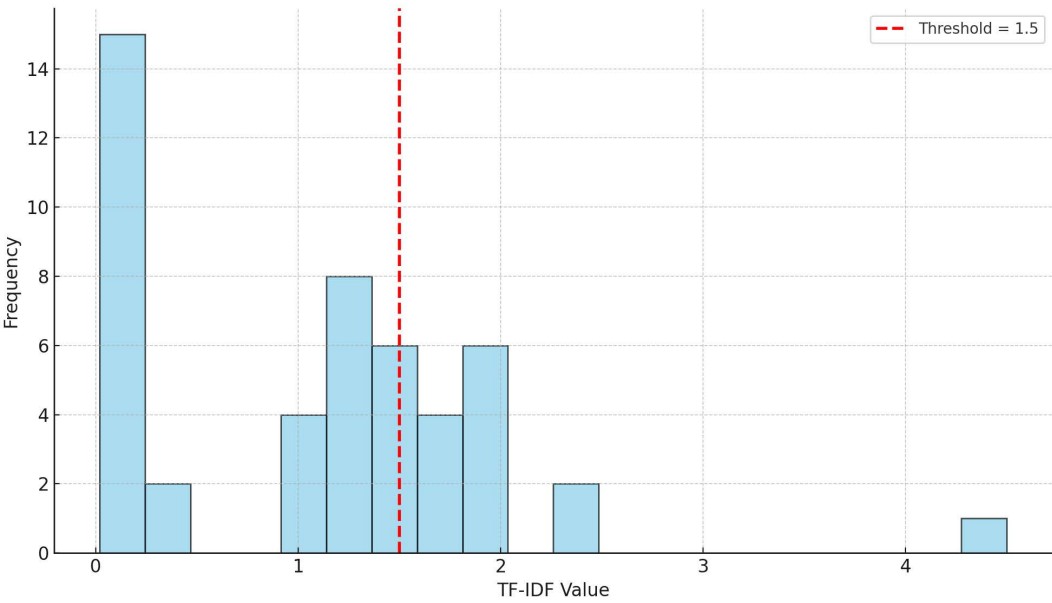

**Fig 2. Histogram of TF-IDF values across fertility-related terms.** Note: The red dashed line marks the dynamic threshold of 1.5 used to identify high-frequency terms for inclusion in semantic clustering. Terms below this threshold were excluded to enhance analytical precision and reduce noise from diffusely distributed vocabulary.

## 2.4. Statistical analysis

Chi-square tests were used to examine associations between categorical variables (e.g., age, marital status) and fertility intentions, and to assess the distribution of high-frequency terms identified through TF-IDF across sentiment categories. Collinearity diagnostics were conducted before regression analysis, and only variables with minimal multicollinearity were included in the final multinomial logistic model. This ensured model stability and reliable identification of significant socio-demographic predictors of fertility intentions.

## 3. Results

### 3.1 Demographic differentiation of fertility intentions

Among the 11,418 female participants, fertility attitudes were overwhelmingly negative: 71.0% expressed reluctance toward childbearing, 28.5% remained neutral, and only 0.5% indicated positive intentions (Table 2). The limited positive group (n = 52) exhibited distinct demographic features—primarily married, aged 20–30, with 46.3% from Hunan Province and one-third explicitly linking reproductive willingness to the availability of social support.

Chi-square analyses revealed significant associations between fertility intentions and all major demographic variables (all p < 0.001), highlighting clear attitudinal stratification by marital status, age, education, and region. These relationships are visually summarized in Fig 3, which presents the distribution of fertility intentions across key demographic subgroups. Unmarried women displayed the highest level of negative sentiment (82.75%) compared to married women (47.84%, χ² = 1,511.47), suggesting that marital status may partially buffer reproductive pessimism. Age stratification revealed a neutral attitudinal majority among adolescents under 20 (91.49%), but negative sentiment escalated in older groups, reaching 89.23% among women over 30 (χ² = 3,033.88), potentially reflecting cumulative life-course pressures.

Educational attainment showed a progressive inverse relationship with reproductive positivity: negative sentiment increased from 25.3% among less-educated women to 76.9% among bachelor's degree holders and 77.6% among

**Table 2. Social and geographical characteristics, and fertility perspectives of network users.**

| Variables | Fettility Intentions(N,100%) | | | Sum | Percentage | $\chi^2$ | P |
|---|---|---|---|---|---|---|---|
| | Negative | Neutral | Positive | | | | |
| **Marital Status** | | | | | | 1511.47 | <0.001 |
| Married | 1834,47.84% | 1974,51.493% | 26,0.68% | 3834 | 33.58% | | |
| Single | 6276,82.75% | 1282,16.90% | 26,0.34% | 7584 | 66.42% | | |
| **Age Interval** | | | | | | 3033.88 | <0.001 |
| Under 20 | 57,8.09% | 645,91.49% | 3,0.43% | 705 | 6.17% | | |
| 20-30 | 2288,53.81% | 1939,45.60% | 25,0.59% | 4252 | 37.24% | | |
| Above 30 | 5765,89.23% | 672,10.40% | 24,0.37% | 6461 | 56.59% | | |
| **Educational Level** | | | | | | 1602.05 | <0.001 |
| Below Bachelor's | 347,25.3% | 1011, 73.9% | 10,0.7% | 1368 | 12.0% | | |
| Bachelor's | 4165,76.9% | 1242,22.9% | 8,0.1% | 5415 | 47.4% | | |
| Beyond Bachelor's | 3598,77.6% | 1003,21.6% | 34,0.7% | 4635 | 40.5% | | |
| **Residence** | | | | | | 2702.03 | <0.001 |
| Top Tier Cities▷ | 1719, 72.38% | 640, 26.95% | 16, 0.67% | 2375 | 20.80% | | |
| Hunan | 3788, 84.8% | 655, 14.66% | 24, 0.54% | 4467 | 39.12% | | |
| Shandong | 12, 1.8% | 652, 97.75% | 3, 0.45% | 667 | 5.84% | | |
| Tianjin | 353, 34.88% | 651, 64.33% | 8, 0.79% | 1012 | 8.86% | | |
| Hubei | 2238, 77.25% | 658, 22.71% | 1, 0.03% | 2897 | 25.37% | | |
| **Data Posted Years** | | | | | | 488.90 | <0.001 |
| 2021 | 1108, 56.79% | 843, 43.21% | – | 1951 | 17.09% | | |
| 2022 | 1686, 66.96% | 787, 31.25% | 45, 1.79% | 2518 | 22.05% | | |
| 2023 | 2037, 71.75% | 800, 28.18% | 2, 0.07% | 2839 | 24.86% | | |
| 2024 | 3279, 79.78% | 826, 20.1% | 5, 0.12% | 4110 | 36.00% | | |
| **Sum** | 8110, 71.03% | 3256, 28.52% | 52, 1.98% | 11418 | 100.00% | | |

Note:▷Top Tier Cities refers to Beijing, Shanghai, Guangzhou, and Shenzhen.

postgraduates ($\chi^2$ = 1,602.05). Spatial patterns followed similar gradients. First-tier cities, marked by intense socioeconomic pressures, exhibited the most negative skew (72.38%, $\chi^2$ = 2,702.03), while provinces such as Shandong maintained predominantly neutral attitudes (97.75%), indicating regional divergence in reproductive perception.

Across the study period, negative attitudes intensified markedly—from 56.79% in 2021 to 79.78% in 2024—mirroring a concurrent decline in neutral sentiment (from 43.21% to 20.10%; $\chi^2$ = 488.90). This temporal trend reflects a rapid shift toward reproductive pessimism.

### 3.2. Salient lexical patterns in fertility discourse

The textual landscape of fertility intentions was constructed through an integrative computational analysis, combining Chi-square validation with TF-IDF semantic weighting, showed in Table 3. Sixteen recurrent terms were extracted from the upper ranks of TF-IDF scoring, reflecting discourse patterns related to economic conditions, sociocultural norms, psychosocial pressures, and reproductive support structures. Among these, high-frequency entries were identified based on dynamic thresholding and visualized in an attitude distribution heatmap (Fig 4), where chromatic gradients captured sentiment-term co-occurrence intensities across the discourse corpus, independent of but complementary to TF-IDF metrics.

Negative sentiment was most consistently anchored by high-weight terms such as housing cost (TF-IDF = 2.34), career advancement and personal goals (TF-IDF = 2.35), and psychosocial adjustment issues (TF-IDF = 2.00), reflecting

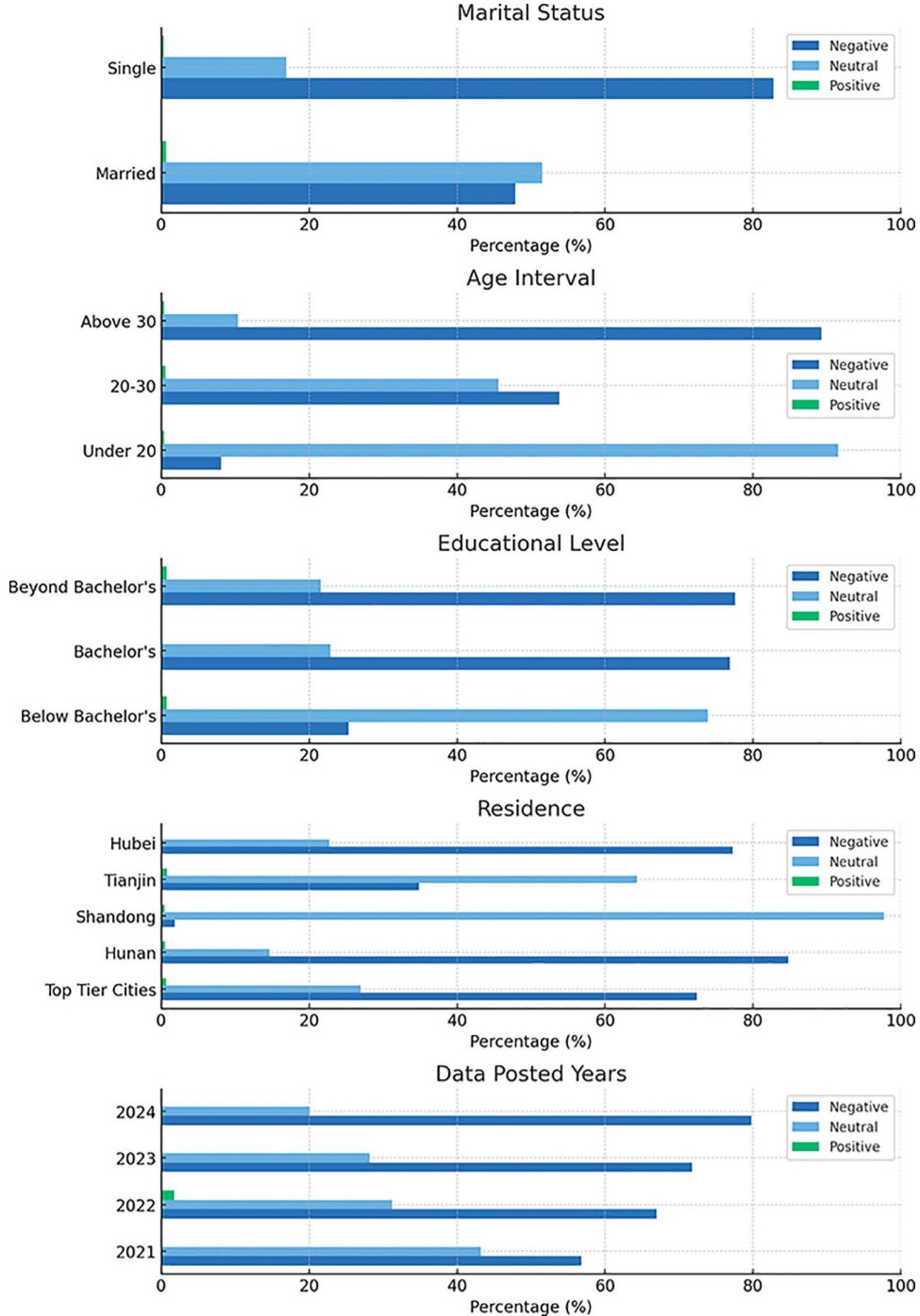

**Fig 3. Distribution of Fertility Intentions by Demographic Subgroups. Note**: The figure illustrates consistent trends of elevated negative sentiment among single, older, and highly educated women, as well as a temporal shift toward reproductive pessimism between 2021 and 2024.

**Table 3. Quantitative Analysis of External Factors Associated with Fertility Intentions.**

| Categories | Extracted Terms | $\chi^2$ | P | TF-IDF (Negative) | TF-IDF (Neutral) | TF-IDF (Positive) |
|---|---|---|---|---|---|---|
| **Socioeconomic Status** | Urban Mobility | 3.60 | <0.001 | 1.23 | 1.49 | 0.02 |
| | Housing cost* | 11.84 | <0.001 | 2.34 | 1.12 | 0.04 |
| | Child-Rearing Expenses* | 2.01 | <0.05 | 0.35 | 4.50 | 0.03 |
| | Adaptation of Urban Newcomers* | 7.93 | <0.05 | 1.78 | 1.35 | 0.05 |
| **Social Support** | Maternal Stress* | 1.96 | 0.05 | 1.02 | 1.65 | 0.06 |
| | Workplace Pregnancy Equity* | 4.01 | <0.01 | 1.56 | 1.43 | 0.09 |
| | Childcare Services* | 3.80 | <0.01 | 1.89 | 1.20 | 0.07 |
| | Career Advancement and Personal Goals* | 4.51 | <0.01 | 2.35 | 1.98 | 0.06 |
| **Social Psychological Factors** | Childbirth Care* | 5.33 | <0.05 | 1.75 | 1.20 | 0.12 |
| | Pregnancy-Related Health Concerns* | 7.12 | <0.01 | 1.88 | 1.30 | 0.15 |
| | Work-Life Balance | 4.20 | <0.01 | 1.45 | 1.10 | 0.08 |
| | Psychosocial Adjustment Issues* | 6.50 | <0.05 | 2.00 | 1.35 | 0.18 |
| **Socio-Cultural Factors** | DINK (Dual Income No Kids)* | 2.87 | <0.05 | 1.90 | 1.45 | 0.05 |
| | Diverse Sexual Orientations (e.g., Lesbian) | 3.10 | <0.05 | 1.15 | 1.20 | 0.20 |
| | Deinstitutionalization Of Marriage* | 6.80 | <0.05 | 1.65 | 1.10 | 0.25 |
| | Social and Environmental Change* | 7.75 | <0.01 | 2.00 | 1.50 | 0.12 |

Table Note:

1. Extracted Terms: Terms were extracted using TF-IDF analysis from UGC on social media platforms. A dynamic threshold of 1.5 was applied to identify high-frequency terms, marked with an asterisk (*). These terms are considered highly relevant to fertility intentions. Terms below the threshold, while still meaningful, provide broader context to the fertility discussion. Definitions of all extracted external factors can be found in S3 Appendix.

2. $\chi^2$: Represents the statistical significance of the relationship between each term and fertility intentions; P: Shows the p-value for each term, with values <0.05 indicating significant relationships.

TF-IDF (Negative), TF-IDF (Neutral), TF-IDF (Positive): TF-IDF scores for each term in relation to negative, neutral, and positive fertility intentions, respectively·

a convergence of material insecurity and individual constraint. Childbirth care (TF-IDF = 1.75) and pregnancy-related health concerns (TF-IDF = 1.88) further highlighted health-related apprehensions, reinforcing a narrative of cumulative vulnerability.

In the neutral sentiment group, child-rearing expenses stood out with the highest TF-IDF score across the corpus (TF-IDF = 4.50), co-occurring with 92.6% of neutral posts. Despite its conventional interpretation as a fertility barrier, its positioning here suggests a more contingent form of hesitation—where financial strain informs postponement rather than rejection. Other moderately weighted terms such as maternal stress (TF-IDF = 1.65) and workplace pregnancy equity (TF-IDF = 1.43) underscore this theme of cost-benefit ambivalence without tipping into pessimism.

Some lexical items presented subtle internal contradictions. For instance, childcare services (TF-IDF = 1.20) appeared more prominently in neutral posts, yet in discourse it often accompanied complaints about systemic inadequacies. Similarly, social and environmental change (TF-IDF = 2.00 in negative, 1.50 in neutral, 0.12 in positive) showed dispersed sentiment distribution, reflecting complex user interpretations of risk and transformation.

Positive sentiment, though rare, was lexically associated with deinstitutionalization of marriage (TF-IDF = 0.25) and diverse sexual orientations (TF-IDF = 0.20), suggesting a distinct set of reproductive values detached from traditional family structures. While statistically marginal, these terms point to emerging minority perspectives embracing alternative fertility ideologies.

Terms with low weight or diffuse sentiment alignment, such as urban mobility (TF-IDF = 1.23), were excluded from further modeling. Overall, the evidence demonstrates that lexical signals of fertility intention cluster meaningfully around

distinct psychological and structural domains, each shaping emotional stance in non-binary and sometimes ambivalent ways.

### 3.3. Multinomial logistic regression analysis

The multinomial logistic regression model, validated through collinearity diagnostics (Tolerance: 0.715–0.957; VIF: 1.045–1.398), quantified the distinct associations between structural, psychological, and temporal variables and fertility intentions, using neutral attitudes as the reference category (Table 4 and Table 5). The effect sizes and confidence intervals, presented in Figs 5 and 6, demonstrate the asymmetrical patterns underlying negative and positive fertility sentiments.

Married women were significantly less likely to express negative intentions (OR = 0.229, 95% CI: 0.194–0.271) and more inclined toward positive responses (OR = 2.756, 95% CI: 1.184–3.716), relative to those with neutral attitudes—suggesting that institutional or economic security continues to play a buffering role. However, the limited expression of positivity even within this group hints at a more fragile effect than traditional frameworks might imply.

Education introduced a bifurcation. Women with less than a bachelor's degree showed an 85% decrease in the likelihood of negative expression (OR = 0.153, 95% CI: 0.124–0.191), but no significant shift toward positivity. Conversely,

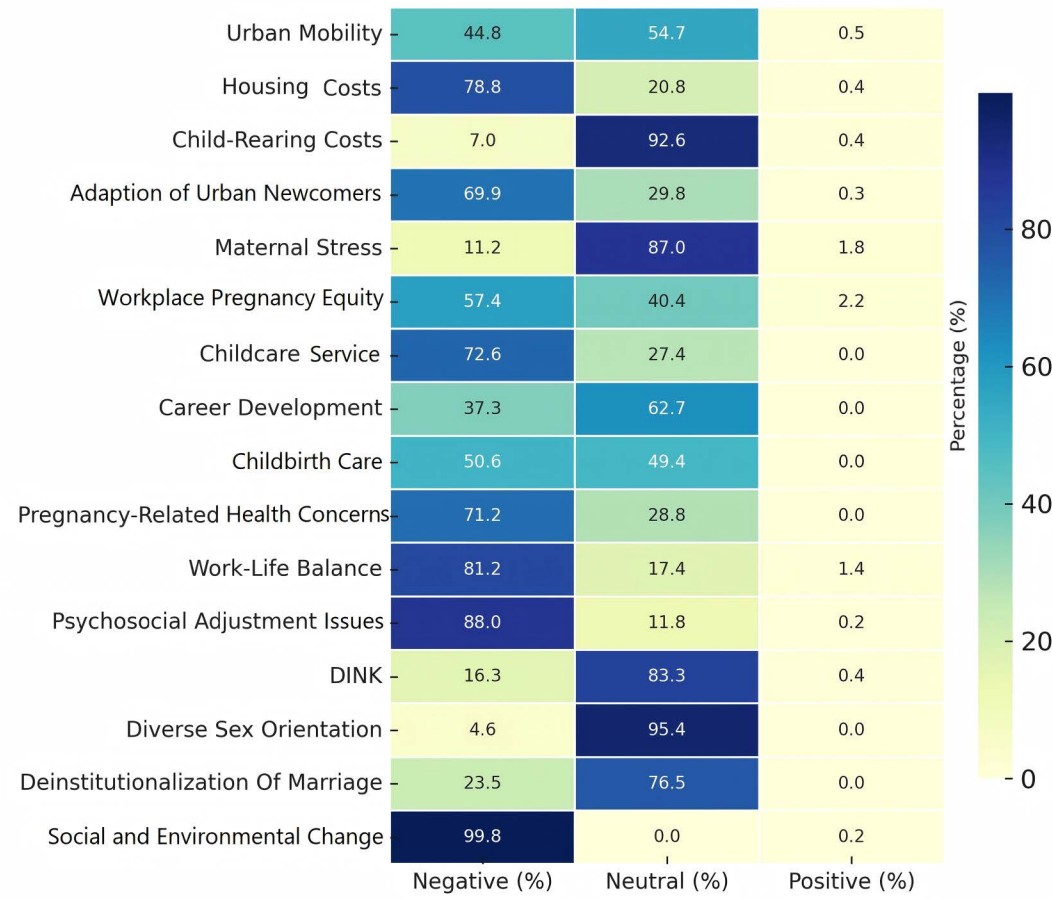

**Fig 4. Percentage Distribution of Fertility Intentions Across External Factors.** Note: Percentage Distribution of Fertility Intentions Across External Factors. This heatmap visualizes how various external factors, such as urban mobility, housing costs, and child-rearing costs, impact fertility intentions, with a clear delineation of negative, neutral, and positive responses based on these influences.

**Table 4. Collinearity Diagnostics for Independent Variables.**

| Cofficient[a] | | | | | | | | |
|---|---|---|---|---|---|---|---|---|
| Variable | B | Beta | t | P | 95% CI | SE | Tolerance | VIF |
| (Constant) | 0.393 | | 16.838 | 0.000 | 0.347 - 0.438 | 0.023 | | |
| Age Range | 0.317 | 0.421 | 45.987 | 0.000 | 0.303 - 0.330 | 0.007 | 0.736 | 1.359 |
| External Factors | 0.042 | 0.092 | 11.143 | 0.000 | 0.035 - 0.049 | 0.004 | 0.911 | 1.097 |
| Marital Status | 0.167 | 0.172 | 18.525 | 0.000 | 0.150 - 0.185 | 0.009 | 0.715 | 1.398 |
| City of Residence | −0.026 | −0.072 | −8.449 | 0.000 | −0.032 - 0.02 | 0.003 | 0.857 | 1.167 |
| Education Level | 0.054 | 0.081 | 9.604 | 0.000 | 0.043 - 0.065 | 0.006 | 0.874 | 1.145 |
| Data Collected Years | 0.039 | 0.1 | 12.494 | 0.000 | 0.033 - 0.045 | 0.003 | 0.957 | 1.045 |

[a]Dependent Variable: Fertility Intention.

postgraduates were nearly three times more likely to express negativity (OR = 2.819, 95% CI: 2.408–3.300) and significantly less likely to report positive views (OR = 0.145, 95% CI: 0.068–0.613), reflecting perceived opportunity costs and possible ideological divergence from normative expectations.

Among affective dimensions, socio-cultural burden exhibited the most polarized pattern: it dramatically increased the odds of negativity (OR = 11.110, 95% CI: 9.070–12.116) while virtually eliminating the chance of positivity (OR = 0.012, 95% CI: 0.003–0.048). Psychological stressors followed a similar trajectory (OR = 8.280 for negative; OR = 0.109 for positive), and socioeconomic pressures also aligned with this pattern, albeit more moderately (OR = 1.352 for negative; OR = 0.249 for positive).

Regional effects further reinforced these divides. Women in top-tier cities were four times more likely to report negative intentions (OR = 4.051, 95% CI: 3.286–4.995), and 95% less likely to report positive ones (OR = 0.046, 95% CI: 0.017–0.124), compared to those in Hubei. A similar pattern appeared in Hunan (OR = 2.127 for negative; OR = 0.042 for positive), while Tianjin showed a notable reduction in negative intent (OR = 0.241, 95% CI: 0.189–0.307) but no corresponding uptick in positive orientation.

Age patterns revealed a clear life-course gradient. Women under 20 were largely absent from both ends of the spectrum (OR = 0.012 for negative; OR = 0.073 for positive), suggesting delayed engagement with fertility decisions. In contrast, those aged 20–30 exhibited a strong positive orientation (OR = 4.250, 95% CI: 2.120–8.460) and a notably reduced likelihood of negative responses (OR = 0.109, 95% CI: 0.092–0.129), identifying them as a key transitional group.

Temporal shifts suggest rising pessimism. Compared to 2024, women surveyed in 2023 were significantly less likely to express negativity (OR = 0.171, 95% CI: 0.138–0.211) and far more likely to report positive intentions (OR = 81.060, 95% CI: 70.338–85.148). This optimism was also evident in 2021 (OR = 16.166 for positive), but absent in 2022, which showed a moderate decline in negative sentiment (OR = 0.699) without a corresponding rise in positivity (p = 0.155).

Collectively, these findings highlight that fertility intentions are not simply attitudinal gradients but structurally segmented orientations. Negative responses reflect widespread yet diffuse suppression, while positive attitudes are confined to narrower ideological and demographic zones. These outcomes are not mirror opposites, but reflect distinct logics shaped by social context.

## 4. Discussion

This study presents a comprehensive examination of how fertility intentions among urban Chinese women are shaped and articulated amidst escalating structural pressures. Over the study period, reproductive pessimism increased by more than 20 percentage points, indicating a growing gap between reproductive aspirations and perceived feasibility. Rather than indicating a mere decline in fertility desire, this trend suggests a broader pattern of attitudinal disengagement that arises

**Table 5. The Multinomial Logistic Regression Model.**

| Fertility intentions[a] | | B | S.D | P | OR | 95% CI | |
|---|---|---|---|---|---|---|---|
| | | | | | | lower | upper |
| **Negative** | intercept | 3.209 | 0.131 | <0.001 | | | |
| | **Marrital status** | | | | | | |
| | Married | −1.472 | 0.085 | <0.001 | 0.229 | 0.194 | 0.271 |
| | Single(ref) | | | | | | |
| | **Education** | | | | | | |
| | Below Bachelor's | −1.874 | 0.111 | <0.001 | 0.153 | 0.124 | 0.191 |
| | Above Bachelor's | 1.036 | 0.08 | <0.001 | 2.819 | 2.408 | 3.3 |
| | Bachelor's(ref) | | | | | | |
| | **Affect factors** | | | | | | |
| | Socio-cultural Factors | 2.407 | 0.131 | <0.001 | 11.11 | 9.07 | 12.116 |
| | Socioeconomic Status | 0.302 | 0.099 | 0.002 | 1.352 | 1.114 | 1.641 |
| | Social psychological factors | 2.114 | 0.125 | <0.001 | 8.28 | 7.095 | 9.154 |
| | Social Support(ref) | | | | | | |
| | **Residence** | | | | | | |
| | Tianjin | −1.423 | 0.124 | <0.001 | 0.241 | 0.189 | 0.307 |
| | Top tier cities | 1.399 | 0.107 | <0.001 | 4.051 | 3.286 | 4.995 |
| | Hunan | 0.755 | 0.097 | <0.001 | 2.127 | 1.759 | 2.572 |
| | Shandong | −4.645 | 0.329 | <0.001 | 0.0097 | 0.005 | 0.318 |
| | Hubei(ref) | | | | | | |
| | **Age Interval** | | | | | | |
| | <20 | −4.437 | 0.18 | <0.001 | 0.012 | 0.008 | 1.017 |
| | 20-30 | −2.217 | 0.086 | <0.001 | 0.109 | 0.092 | 0.129 |
| | >30(ref) | | | | | | |
| | **Data Collected Years** | | | | | | |
| | 2021 | 0.147 | 0.103 | 0.153 | 1.159 | 0.947 | 1.418 |
| | 2022 | −0.358 | 0.107 | 0.001 | 0.699 | 0.566 | 0.863 |
| | 2023 | −1.766 | 0.108 | <0.001 | 0.171 | 0.138 | 0.211 |
| | 2024(ref) | | | | | | |
| **Positive** | intercept | −5.405 | 1.192 | <0.001 | | | |
| | **Marrital status** | | | | | | |
| | Married | 1.014 | 0.347 | 0.003 | 2.756 | 1.184 | 3.716 |
| | Single(ref) | | | | | | |
| | **Education** | | | | | | |
| | Below Bachelor's | 0.291 | 0.498 | 0.558 | 0.747 | 0.282 | 1.982 |
| | Above Bachelor's | −1.934 | 0.415 | <0.001 | 0.145 | 0.068 | 0.613 |
| | Bachelor's(ref) | | | | | | |
| | **Affect factors** | | | | | | |
| | Socio-cultural Factors | −4.401 | 0.699 | <0.001 | 0.012 | 0.003 | 0.048 |
| | Socioeconomic Status | −1.389 | 0.429 | 0.001 | 0.249 | 0.108 | 0.578 |
| | Social psychological factors | −2.218 | 0.427 | <0.001 | 0.109 | 0.047 | 0.251 |
| | Social Support(ref) | | | | | | |
| | **Residence** | | | | | | |
| | Tianjin | 1.782 | 1.074 | 0.097 | 5.94 | 0.724 | 6.742 |
| | Top tier cities | −3.089 | 1.057 | 0.003 | 0.046 | 0.017 | 0.124 |
| | Hunan | −3.165 | 1.034 | 0.002 | 0.042 | 0.024 | 0.582 |

*(Continued)*

**Table 5.** (Continued)

| Fertility intentions[a] | | B | S.D | P | OR | 95% CI | |
|---|---|---|---|---|---|---|---|
| | | | | | | lower | upper |
| | Shandong | 0.311 | 1.193 | 0.794 | 1.365 | 0.132 | 1.151 |
| Hubei(ref) | | | | | | | |
| **Age Interval** | | | | | | | |
| | <20 | −2.62 | 0.66 | <0.001 | 0.073 | 0.02 | 0.265 |
| | 20-30 | 1.447 | 0.345 | <0.001 | 4.25 | 2.12 | 8.46 |
| >30(ref) | | | | | | | |
| **Data Collected Years** | | | | | | | |
| | 2021 | 2.783 | 0.506 | <0.001 | 16.166 | 6.002 | 18.547 |
| | 2022 | −1.242 | 0.874 | 0.155 | 0.289 | 0.052 | 1.602 |
| | 2023 | 4.396 | 2.18 | 0.004 | 81.06 | 70.338 | 85.148 |
| | 2024(ref) | | | | | | |

from the cumulative effects of economic, institutional, cultural, and ecological pressures. Framed within the cognitive-social model, these intentions emerge as adaptive responses to the dynamic interplay between internal evaluations and external structural limitations. They represent not merely personal reluctance but also a broader cognitive and emotional recalibration of reproductive decisions in the context of modern urban environments.

Economic insecurity emerged as a consistently salient factor associated with negative fertility attitudes, particularly in relation to gendered labor dynamics and institutional constraints—a pattern further contextualized through observed discourse on educational stratification, workplace inequality, and social support inadequacy. Housing costs (TF-IDF = 2.34) and child-rearing expenses (TF-IDF = 4.50) were consistently identified as central financial stressors across discourse and regression models. While 33 Chinese cities surpassed the World Bank's high-income GDP threshold by 2023 [39]—this concentration of wealth has intensified cost-of-living pressures. Paradoxically, economic advancement exacerbates reproductive constraints by amplifying financial thresholds for family formation. Housing unaffordability—fueled by population concentration, speculative markets, and land scarcity—serves not only as a material obstacle but as a symbolic deterrent [40–43]. A Beijing mother's remark that her ¥7.2 million mortgage constitutes "not just debt but a 30-year sentence that precludes children's futures" vividly captures the symbolic weight of homeownership in China's post-reform social contract. Property ownership functions not only as a material asset but increasingly as a perceived prerequisite for intergenerational stability. This reality is further compounded by the substantial financial burden of raising a child in China, estimated at approximately 680,000 yuan from birth to university graduation [44]. Such pressures place significant strain on women balancing family planning with economic stability. These dynamics mirror patterns in other low-fertility societies, including Italy and Norway, where economic affluence paradoxically coexists with widespread reproductive pessimism [45,46].

Age and educational attainment interact to deepen the structural disincentives surrounding fertility decisions. As women pursue higher education, the extended duration of academic engagement delays entry into reproductive life stages, deferring marriage and parenthood. Beyond postponement, this trajectory increases exposure to work environments where the opportunity costs of motherhood grow more pronounced [47]. Regression results confirm that both increasing age and higher educational attainment are associated with greater likelihood of negative fertility intentions and diminished positive orientation—a trend consistent with patterns observed across other low-fertility contexts [48,49].For highly educated women, transitioning into the labor market often brings intensified exposure to institutional barriers. In China, recent evidence highlights heightened work–family conflicts that lead many women to postpone or forgo parenthood altogether [50, 51]. Discourse-level analysis supports this: TF-IDF scores for terms like "career advancement" (2.35) and "workplace pregnancy equity" (1.56) reflect sustained anxiety over hiring discrimination, career stagnation, and long-term

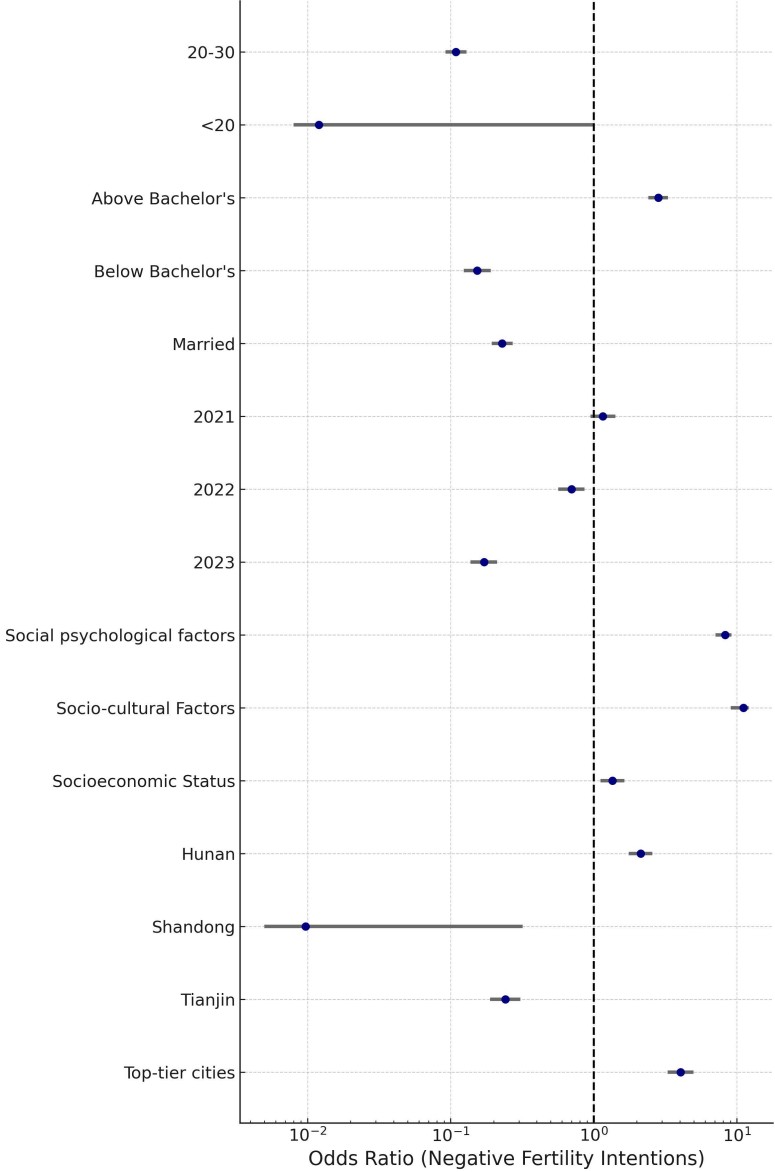

**Fig 5. Adjusted Predictors of Negative Fertility Intentions.**

wage penalties. These findings are echoed in international literature indicating that in the absence of institutional support, motherhood is increasingly framed as a high-risk, low-reward decision, particularly for professional women [52].

Workplace norms and policy infrastructures critically mediate this calculus. Nordic countries have historically offered generous support—universal childcare, gender-equal parental leave—which enables 89% maternal labor market reentry and reduces work–family tension [53]. However, recent fertility declines in the region suggest that institutional generosity alone may not be sufficient to sustain reproductive optimism. In contrast, China's fragmented childcare system and minimal paternal leave—averaging just 15 days, with two-thirds of fathers taking fewer than five—continue to reinforce the male-breadwinner and female-caregiver model [54,55]. These systemic gaps leave women to absorb disproportionate burdens during the perinatal period and beyond, particularly in high-pressure professional settings. The resulting conflict

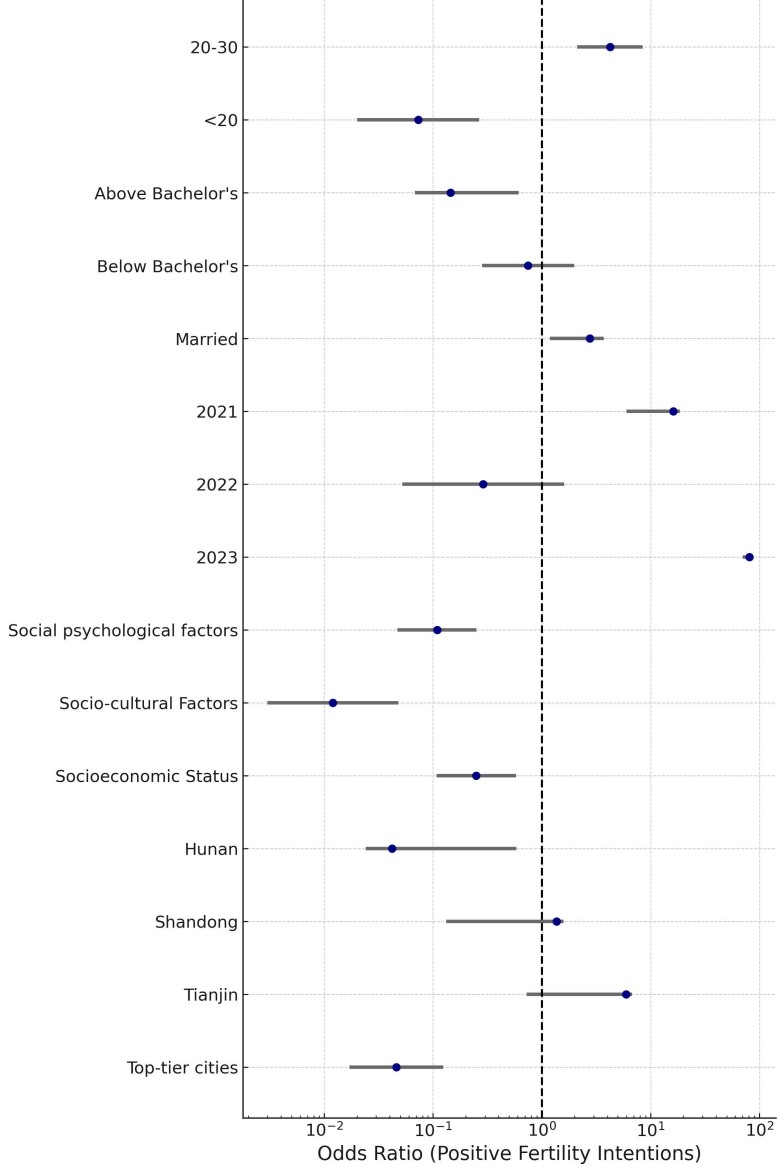

**Fig 6. Adjusted Predictors of Positive Fertility Intentions. Note**: Forest plot illustrating the results of the multinomial logistic regression model, with neutral fertility intention as the reference category. Each horizontal line represents a 95% confidence interval for a given predictor, and the central marker denotes the adjusted odds ratio (OR). ORs > 1 indicate an increased likelihood of expressing either negative or positive fertility sentiment relative to neutral intention, while ORs < 1 reflect a reduced likelihood. The model was adjusted for multicollinearity (VIF < 1.4), and only statistically significant predictors (p < 0.05) are displayed.

between career advancement and childbearing reflects not only personal trade-offs but broader institutional inertia, reinforcing structural gender inequality across both public and private domains.

Institutional pressures also extend into the private sphere. Marriage continues to influence fertility intentions but in more complex ways than traditionally assumed. While regression results indicated that married women were less likely to express negative intentions (OR = 0.229) and more likely to express positive ones (OR = 2.756), the broader discourse reveals a striking divergence from this statistical trend. Even among married women, positive sentiment was notably

subdued, suggesting that the institution of marriage—despite its traditionally protective role in fertility, as frequently emphasized in demographic literature [23,41]—no longer guarantees emotional affirmation or perceived readiness for parenthood. This disconnect reflects a growing skepticism toward the protective value of marriage, particularly in contexts where women continue to bear a disproportionate share of domestic responsibilities. According to national surveys, married women in China perform on average 1.35 more hours of housework per day than men and account for 73% of all childcare responsibilities [56–58]—an embodiment of the "second shift," where women shoulder a dual burden of paid and unpaid labor within the household [59]. This imbalance is not merely symbolic; it manifests in tangible material penalties—most notably, a 49% earnings reduction post-childbirth, a disparity absent among men and largely driven by prolonged career interruptions, reduced working hours, and limited promotion opportunities stemming from caregiving responsibilities [60]. In this light, marriage amplifies rather than mitigates reproductive risk. Its institutional formality offers symbolic security, but little practical redistribution of risk or care.

Cultural reconfigurations are simultaneously challenging normative expectations of reproduction. The analysis identified growing discourse around DINK lifestyles, voluntary childlessness, and the retreat from marriage as a reproductive institution—particularly among younger, educated urban women. These choices are increasingly presented as rational, empowered responses to a precarious and uncertain world, reflecting Inglehart's post-materialist value thesis that prioritizes self-actualization over traditional family obligations [61]. Terms like "DINK" (TF-IDF = 1.90) and "deinstitutionalization of marriage" (TF-IDF = 1.65) demonstrated significant salience in negative intention discourse, supported by regression results linking socio-cultural framing to an elevenfold increase in negative sentiment (OR = 11.110) and a sharp decline in positive expression (OR = 0.012). References to "self-partnered" life choices (e.g., Xiaohongshu user: "I have no need for a wedding to validate my life choices") reflect the normalization of reproductive nonconformity. These trajectories, consistent with findings in Western societies [62, 63], often represent a pursuit of personal freedom and relational autonomy rather than outright rejection of cultural norms. Importantly, such narratives are disproportionately voiced by individuals who do not cite financial barriers, implying that higher socio-economic capital enables greater latitude to opt out of reproductive expectations. This discursive evolution is thus both ideational and stratified: agency is not uniformly distributed but shaped by access to resources, education, and relational independence.

A novel dimension emerged in the form of ecological anxiety. "Social and environmental change" (TF-IDF = 2.00) frequently co-occurred with negative attitudes, accompanied by expressions such as "this world is a death sentence for the next generation." These sentiments, aligned with Schneider-Mayerson and Leong's theory of eco-reproductive ethics, are grounded in lived concerns about climate instability, environmental degradation, and global political uncertainty. Unlike economic pressures, ecological anxieties are more ideologically encoded and largely absent in positive discourse, suggesting a deeply moralized framing of reproductive restraint. This asymmetric semantic distribution reinforces that while negative intentions are multidimensional, positive ones tend to cluster around a narrower band of structural security. Recent systematic reviews further underscore this trend, positioning voluntary childlessness as a responsible ethical choice in the face of climate crisis [64,65]. In China's densely populated urban centers, such expressions reflect a psychological burden that transcends material calculations, underscoring the growing moralization and existential framing of fertility decisions within the Anthropocene context.

Taken together, these findings reveal a fundamental misalignment between individual reproductive capacities and the institutional, economic, and sociocultural conditions in which reproductive decisions unfold. Fertility intentions—framed through cognitive–social, behavioral, and motivational lenses—are not isolated reflections of personal desire, but adaptive responses to systemic constraints that increasingly render parenthood a high-risk endeavor. It is both analytically reductive and ethically problematic to interpret declining fertility as the failure of individual women; rather, it reflects a rational recalibration in the face of unmet structural support and institutional inertia. This shift marks a broader trajectory in human development, from biologically driven reproduction toward a deliberate pursuit of autonomy, well-being, and moral agency. As material security expands, individuals prioritize reflective life planning over normative imperatives, reshaping fertility

as a matter of choice rather than obligation. In this context, fertility decline should not be pathologized as a demographic crisis, but recognized as a transitional expression of modernity—one that redefines human value beyond reproduction toward equity, sustainability, and self-determination. Responding to this transformation requires not only structural reform—gender-equitable labor protections, universal childcare, and inclusive parental leave—but also a cultural realignment that affirms reproductive autonomy as a socially supported, ethically grounded, and individually empowered choice.

## 5. Limitations

This study provides insight into fertility intentions among urban Chinese women through large-scale user-generated content, yet several methodological and interpretive limitations remain.

The sample is inherently selective, shaped by the demographics of active social media users—typically younger, urban, and more educated—thereby limiting the generalizability of findings beyond this population segment. The cross-sectional nature of the data also restricts inference on long-term intention trajectories or behavioral outcomes. Socio-demographic attributes were inferred through linguistic markers rather than self-report, introducing classification uncertainty despite multi-stage validation procedures. Additionally, the use of culturally specific metaphors and figurative language may obscure semantic clarity in ways not easily captured by computational models. Sentiment classification, while implemented through an established neural network system (IBM Watson NLU), reflects the interpretive constraints of its training corpus. Outputs should be understood as probabilistic estimations rather than objective representations of affective stance. Particularly in high-context languages such as Chinese, rhetorical nuance and moral framing may challenge sentiment polarity schemes.

Although the methodological framework employed here allows for consistent large-scale analysis, caution is warranted when interpreting affective or demographic patterns as definitive. Future research would benefit from incorporating longitudinal data, offline triangulation, and culturally grounded lexical resources to enhance analytical depth and reliability.

## 6. Conclusion

Fertility intentions among urban Chinese women are increasingly shaped not by isolated personal preference, but by how society allocates risk, opportunity, and care. This shift reveals deeper tensions between demographic expectations and the lived conditions in which life decisions unfold. Addressing this disconnect is not only a matter of reversing fertility decline, but of reimagining what social systems must provide to make parenthood a viable and valued choice.

## Supporting information

**S1 Appendix. De-identified Data for Fertility Intentions.** Anonymized dataset of female user-generated content with extracted socio-demographic variables used for analysis.
(XLSX)

**S2 Appendix. TF-IDF Validation and Sentiment Alignment.** Validation of term selection and sentiment classification, including expert ratings and statistical robustness checks.
(PDF)

**S3 Appendix. Definitions of External Factors.** Detailed descriptions and sentiment labels of external factors influencing fertility intentions, grouped by thematic category.
(PDF)

## Author contributions

**Conceptualization:** Jing Xiang.

**Funding acquisition:** Xuan Sun.

**Investigation:** Jing Xiang, Xuan Sun.

**Methodology:** Jing Xiang.

**Supervision:** Xuan Sun.

**Validation:** Jing Xiang.

**Writing – original draft:** Jing Xiang.

**Writing – review & editing:** Jing Xiang, Xuan Sun.

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
