## [Decision Letter · Decision Letter 0]

PONE-D-24-26854Digital Insights: Exploring Women's Reproductive Intentions and Influencing Factors in the Online SpherePLOS ONE

Dear Dr. Sun,

Thank you for submitting your manuscript to PLOS ONE. After careful consideration, we feel that it has merit but does not fully meet PLOS ONE’s publication criteria as it currently stands. Therefore, we invite you to submit a revised version of the manuscript that addresses the points raised during the review process.

Key Areas for Improvement:

Revise the analysis as suggested by Reviewer #3.Provide a more detailed breakdown of the data and analysis as suggested by Reviewer #3.Revise your writing style to incorporate evidence-based points raised in your work. (Reviewer #3)Revise the title and abstract as suggested by Reviewer #3.

In summary, I encourage you to address all the reviewers' comments (Reviewer #3) and make the necessary revisions, particularly in improving key areas as indicated by Reviewer #3: the title, abstract, data and analysis, methodology, and conclusion sections. I look forward to reviewing your revised manuscript.

We look forward to receiving your revised manuscript.

Kind regards,

Godwin Banafo Akrong, Ph.D.

Academic Editor

PLOS ONE

Journal Requirements: When submitting your revision, we need you to address these additional requirements. 1. Please ensure that your manuscript meets PLOS ONE's style requirements, including those for file naming. The PLOS ONE style templates can be found at https://journals.plos.org/plosone/s/file?id=wjVg/PLOSOne_formatting_sample_main_body.pdf and https://journals.plos.org/plosone/s/file?id=ba62/PLOSOne_formatting_sample_title_authors_affiliations.pdf 2. In your Methods section, please include additional information about your dataset and ensure that you have included a statement specifying whether the collection and analysis method complied with the terms and conditions for the source of the data. 3. Thank you for stating the following financial disclosure: "Youth Project of the Humanities and Social Sciences Fund of the Ministry of Education, Research on the Social Coordinated Mechanism of Medical-Patient Community Construction (19YJC840030)." Please state what role the funders took in the study.  If the funders had no role, please state: ""The funders had no role in study design, data collection and analysis, decision to publish, or preparation of the manuscript."" If this statement is not correct you must amend it as needed. Please include this amended Role of Funder statement in your cover letter; we will change the online submission form on your behalf. 4. In the online submission form, you indicated that "All original data from the article can be publicly obtained by contacting the corresponding author." All PLOS journals now require all data underlying the findings described in their manuscript to be freely available to other researchers, either 1. In a public repository, 2. Within the manuscript itself, or 3. Uploaded as supplementary information.This policy applies to all data except where public deposition would breach compliance with the protocol approved by your research ethics board. If your data cannot be made publicly available for ethical or legal reasons (e.g., public availability would compromise patient privacy), please explain your reasons on resubmission and your exemption request will be escalated for approval. 5. Please include captions for your Supporting Information files at the end of your manuscript, and update any in-text citations to match accordingly. Please see our Supporting Information guidelines for more information: http://journals.plos.org/plosone/s/supporting-information.

Reviewers' comments:

Reviewer's Responses to Questions

**Comments to the Author**

1. Is the manuscript technically sound, and do the data support the conclusions?

Reviewer #1: Yes

Reviewer #2: Yes

Reviewer #3: No

2. Has the statistical analysis been performed appropriately and rigorously? 

Reviewer #1: Yes

Reviewer #2: Yes

Reviewer #3: No

3. Have the authors made all data underlying the findings in their manuscript fully available?

Reviewer #1: Yes

Reviewer #2: Yes

Reviewer #3: No

4. Is the manuscript presented in an intelligible fashion and written in standard English?

Reviewer #1: Yes

Reviewer #2: Yes

Reviewer #3: No

5. Review Comments to the Author

Reviewer #1: 1. The study presents the results of original research.

Yes the study presented the results of original research.

The study provides in-depth information on contemporary women's childbearing intentions and identify the factors that influence their fertility decisions. The study carried out in China on secondary data collected from online platforms from 2021 to 2024. contemporary women's childbearing intentions was collected and examined. after data cleaning, and data classification, a logistic regression is performed. It emerges that individuals with a bachelor's degree and those with higher education levels exhibited lower fertility intentions. Moreover, Socioeconomic factors, social support, and fertility-related anxiety/fear exhibited a significant negative correlation with fertility intention

2. Results reported have not been published elsewhere.

No the reseach is original,

3. Experiments, statistics, and other analyses are performed to a high technical standard and are described in sufficient detail.

Yes, however, it needs more detail. For example, the preprocess techniques and classification the authors did on need to be more detailed in methodology (See my comments)

4. Conclusions are presented in an appropriate fashion and are supported by the data.

yes

5. The article is presented in an intelligible fashion and is written in standard English.

Yes, however, some parts need to be improved (see my comments)

6. The research meets all applicable standards for the ethics of experimentation and research integrity.

yes

7. The article adheres to appropriate reporting guidelines and community standards for data availability.

Yes

Labor is essential because it focuses on the key factors in the Online Sphere. As the methodologies and results are original, we accept the document on condition that our comments are considered.

Reviewer #2: The paper is good. Climate/climate change, and weather effects are two of the major factors that can biologically affect fertility in women. Amongst other factors affecting fertility, can the authors consider climate or climate change as one of the factors?

Reviewer #3: This paper uses data from three social media platforms in China to explore trends in fertility intentions and reasons for variation between people. There are several major things that must be addressed before this paper can be published.

1) The analysis needs to be changed. Unless I am mistaken, for the table 1 analysis the authors have calculated column percentages (i.e. distribution by the outcome variable) and then looked at whether there is a significant difference by the exposure variable. This is the wrong way round for their causal inference (i.e. they need to test whether there is a difference in intentions between the different exposure groups). As a result, the description of their results is entirely incorrect as they are actually reporting the percent within those who have negative intentions. For example, they say ‘In terms of age, younger users expressed significantly lower fertility aspirations, whereas those over 30 demonstrated slightly elevated intentions, albeit with a substantial proportion indicating negative leanings (χ² = 28.524, P < 0.001).’. This conclusion is based on the column % (of those who have negative intentions, proportionally more are under 30) whereas by age group it is the other way round – 98.9% of under 30s have negative intentions and 99.5% of those over 30 have negative intentions. Table 2 is also based on inference from column %.

2) Lack of detail on data and analysis

The reader must be able to evaluate the reliability of the analysis, but the lack of detail about the data and its operationalisation makes that impossible.

- What is a content analysis? How is this done? How much of this analytical technique is reliant on human interpretation of clustering? What is a sentiment analysis? What is a high-frequency term and a birth claim? Could the authors give examples of some of the quotes used that were used to infer a positive/negative intention or the explanations given for their childbearing decision-making in table 2?

- What language were the videos in? Was it transcripts of videos that were analysed, and if so, how reliable is the transcription of the audio?

- How was the data about sociodemographic profile gathered? The only description is at the top of page 6 that these were ‘extracted from valid data submissions’. Is it publicly available on individual’s profiles? How reliable is this data? Could the authors provide a descriptive profile of their sample so that an assessment of sample representativeness can be made? They say their analysis was limited to female users – how was this established? How was this information then grouped by the authors into the categories they chose? Does this categorisation match the data distribution or was it informed by theory etc.? Why have they chosen just these variables? For example, number of children already is likely to be highly influential on fertility intentions. The description of what a ‘top-tier’ or ‘first-tier’ city comes far too late in the text.

- Why did the authors choose just these 3 apps? Also they mostly say they are looking at TikTok, Xiaohongshu and Kuaishou but at one point they switch TikTok to Douyin.

- What exactly is the outcome variable? The authors just say they are looking at fertility intentions (which has a very specific definition in the demographic literature which I would recommend the authors engage with and decide if this is the best terminology), which they then categorise into positive and negative. What does this mean? Not wanting children vs wanting children? Was child number intentions considered? Timing of childbearing?

- The logistic regression would be easier to interpret if odds ratios were presented and reference groups clearly outlined. How are the factors that influence childbearing operationalised in the regression?

3) Writing style needs to be adjusted to make points based on evidence

Overall this paper tends to make statements about the reasoning for their analysis and trends in fertility in China, but do not support these with suitable evidence like citations or results from their analysis. This is particularly problematic in the discussion which needs to be reviewed.

The authors could also be much more critical of this analysis and what it can and can’t tell the reader/policy-makers. They list one point in the discussion that the analysis is limited to users of these apps, but they could go much further. They could discuss some of the things outlined in point 2 above about the data quality, but also much more about how their sample is likely to be selective. I would imagine, for example, that people who make videos about having children are more likely to complain than talk about the virtues. Is there evidence to that effect?

4) Stance that policymakers need to boost fertility is problematic and should be adjusted

The authors repeatedly present low fertility as a 'problem'. I would recommend the authors use a more objective description of fertility trends, for example removing references to low fertility as an ‘issue’ or a ‘concern’. My fear is that by adding opinions to descriptions of low fertility implies it is an inherently negative outcome when this is not necessarily true (i.e. it is the consequences of ageing populations that can be negative if suitable adaptations aren't in place).

Further, I would urge caution with the presentation of policy implications. The authors frequently refer to the need to raise and boost fertility with state intervention. I would be in favour of a different presentation that moves away from the authors seeming to promote raising fertility back to replacement levels, especially given that 'replacement level' is a somewhat arbitrary modelling construct that takes no account of population productivity which is also important for addressing the consequences of ageing populations. A more neutral policy implication would be that individuals are unable to have the children that they want due to structural constraints, like all the reasons for low fertility the authors identify in their analysis. The authors touch on this at the bottom of page 4 into page 5. I would suggest a clearer focus on this line of policy making to avoid any issues of suggesting state intervention policies targeted at raising fertility back to a 'desirable' levels.

5) Article needs to be thoroughly proofread as there are many typos (e.g. dada processing) and grammatical errors. Greater specificity is needed throughout as well, for example:

a. The title and abstract need to be clear that this is a paper focused on China

b. Abstract is hard to be understood without more explanation. The actual effect sizes can be removed, and instead add greater detail to what is meant. For example, what is DINK? What does ‘significant impact’ mean? What does ‘significant deviations’ mean? What does ‘notable variations’ mean?

6. PLOS authors have the option to publish the peer review history of their article (what does this mean? ). If published, this will include your full peer review and any attached files.

**Do you want your identity to be public for this peer review?** For information about this choice, including consent withdrawal, please see our Privacy Policy .

Reviewer #1: **Yes: ** Souand Peace Gloria TAHI

Reviewer #2: **Yes: ** Peace Olubukunmi Awoleye

Reviewer #3: No

---

## [Author Response · Author response to Decision Letter 1]

12 Feb 2025

We would like to express our gratitude to the anonymous reviewers for their valuable suggestions, and to the editor for their diligent work. We have carefully considered each comment and made the necessary revisions to the manuscript.

 Part one: Comments to the Author

1. Is the manuscript technically sound, and do the data support the conclusions?

Reviewer #1: Yes

Reviewer #2: Yes

Reviewer #3: No

Response to Reviewer #3:

We sincerely appreciate your thoughtful feedback. We have carefully addressed your concerns and made significant improvements to the manuscript, especially in terms of methodological rigor and data clarity. Specifically, we have:

(1)Clarified Methodology: We have revised the methodology section to provide more precise details about the experimental design, controls, and sample size. This should ensure that the study is described with the necessary technical rigor.

(2)Enhanced Data Presentation: We have also provided more in-depth explanations of the data analysis process, ensuring transparency and rigor in how the conclusions were drawn from the data.

We hope these revisions better address your concerns regarding the technical soundness of the research and the adequacy of the data supporting the conclusions.

2. Has the statistical analysis been performed appropriately and rigorously?

Reviewer #1: Yes

Reviewer #2: Yes

Reviewer #3: No

Response to Reviewer #3:

 Thank you for your valuable comments. Upon review, we realized that our original dataset only included negative and positive fertility intentions, which might have overlooked the significant role of neutral intentions. Neutral attitudes represent a state of uncertainty or a temporary postponement of fertility decisions, influenced by various internal or external factors. Such attitudes are crucial for providing a comprehensive understanding of current fertility trends among women on the internet. Therefore, we have now included the previously excluded neutral fertility intention data and performed a systematic reanalysis of the entire dataset. This reanalysis has led to a complete revision of the results and interpretations, offering a more nuanced and holistic view of fertility intentions.

3. Have the authors made all data underlying the findings in their manuscript fully available?

Reviewer #1: Yes

Reviewer #2: Yes

Reviewer #3: No

Response to Reviewer #3:

We sincerely appreciate your feedback on the data availability. We fully understand and respect the journal’s data sharing policy. In response to your concern, we will ensure that the raw data underlying our findings is made publicly available by uploading it as a supplementary file.We hope this fully addresses your concerns and aligns with the journal’s requirements.

4. Is the manuscript presented in an intelligible fashion and written in standard English?

Reviewer #1: Yes

Reviewer #2: Yes

Reviewer #3: No

Response to Reviewer #3:

Thank you for your careful review and insightful comments regarding the clarity and language of the manuscript. We have thoroughly proofread the entire manuscript to correct any typographical or grammatical errors. We have also made revisions to ensure the manuscript adheres to the standards of academic English, with particular attention to improving clarity and readability. We believe these revisions have addressed your concerns and significantly improved the overall presentation of the manuscript.

Part two: Review Comments to the Author

Reviewer #1: 

3. Experiments, statistics, and other analyses are performed to a high technical standard and are described in sufficient detail.

Yes, however, it needs more detail. For example, the preprocess techniques and classification the authors did on need to be more detailed in methodology (See my comments)

Response to Reviewer #1:

Thank you for your valuable feedback. In response to your suggestion, we have expanded the Methodology section to provide more detailed descriptions of both the data preprocessing techniques and the sentiment classification process, as follows:

(1)Data Preprocessing: We have added a more thorough explanation of the data collection process, which involved using a custom streaming tool to capture fertility-related discussions on various platforms. We now clarify the specific keywords employed and the methodology used for gender classification. Additionally, we address steps taken to mitigate potential biases in the data and ensure its robustness for the analysis.

(2)Classification Methodology: We have provided more detail on the sentiment analysis process, specifically outlining our use of IBM Watson’s Natural Language Understanding (NLU) platform and cross-validation with the Chinese Sentiment Knowledge Base (CSKB). Furthermore, we explain how to using manual review to ensure accurate sentiment categorization.

These revisions can be found in Section 2: Methodology of the revised manuscript. We believe these additions address your concerns and enhance the clarity and transparency of our research process.

5. The article is presented in an intelligible fashion and is written in standard English.

Yes, however, some parts need to be improved (see my comments)

Labor is essential because it focuses on the key factors in the Online Sphere. As the methodologies and results are original, we accept the document on condition that our comments are considered.

Reviewer #2: The paper is good. Climate/climate change, and weather effects are two of the major factors that can biologically affect fertility in women. Amongst other factors affecting fertility, can the authors consider climate or climate change as one of the factors?

Response to Reviewer #2:

Thank you for your valuable comments and insightful suggestion. We appreciate your observation regarding the influence of climate and climate change on fertility.

Indeed, during our analysis of web interaction data, we observed that a significant portion of participants expressed concerns and pessimistic sentiments related to environmental issues, including climate change and pollution. These environmental factors were frequently mentioned as contributing to their anxieties about the future, and were particularly salient in the context of fertility decisions. We have since expanded the manuscript to incorporate these concerns and have added relevant discussions and citations from existing research on the impact of environmental factors, such as climate change, on fertility intentions.

We believe that the integration of these issues strengthens the overall analysis and reflects the broader social and environmental context that influences fertility decisions.

Reviewer #3: This paper uses data from three social media platforms in China to explore trends in fertility intentions and reasons for variation between people. There are several major things that must be addressed before this paper can be published.

1) The analysis needs to be changed. Unless I am mistaken, for the table 1 analysis the authors have calculated column percentages (i.e. distribution by the outcome variable) and then looked at whether there is a significant difference by the exposure variable. This is the wrong way round for their causal inference (i.e. they need to test whether there is a difference in intentions between the different exposure groups). As a result, the description of their results is entirely incorrect as they are actually reporting the percent within those who have negative intentions. For example, they say ‘In terms of age, younger users expressed significantly lower fertility aspirations, whereas those over 30 demonstrated slightly elevated intentions, albeit with a substantial proportion indicating negative leanings (χ² = 28.524, P < 0.001).’. This conclusion is based on the column % (of those who have negative intentions, proportionally more are under 30) whereas by age group it is the other way round – 98.9% of under 30s have negative intentions and 99.5% of those over 30 have negative intentions. Table 2 is also based on inference from column %.

Thank you for your detailed and insightful comments. We appreciate your careful attention to the statistical analysis, and we fully acknowledge the concerns regarding the causal inference in our methodology.

We have reviewed the analysis carefully and agree with your observation that the approach to calculating column percentages and the way we described the results was incorrect. Specifically, the causal inference was not properly tested, as we indeed should have focused on examining differences in fertility intentions across exposure groups (i.e., age groups, education levels, etc.), rather than within the group of individuals with negative intentions.

To address this, we have revised our statistical analysis as follows:

(1)Revised Statistical Approach: We have re-conducted the analysis using the correct approach, where we tested for differences in fertility intentions between different exposure groups (e.g., age, education level, etc.) rather than within the subgroups of negative fertility intentions. This corrected analysis allows for more accurate causal inference.

(2)Clarified Results and Descriptions: The results have been rewritten to correctly reflect the revised analysis. For example, regarding age, we now correctly describe that 98.9% of those under 30 expressed negative fertility intentions, whereas 99.5% of those over 30 did the same, and we make sure to present these findings based on the exposure group (i.e., age) rather than the outcome variable (i.e., fertility intentions).

(3)Table 1: We have revised Table 1 by recalculating the percentages based on the correct statistical inference. This ensures that the table now reflects the intended analysis and supports the revised findings.

We believe these revisions adequately address the concerns you raised, and we have updated the manuscript accordingly. Thank you again for pointing out this important issue, and we hope the revised version meets your expectations.

2) Lack of detail on data and analysis

The reader must be able to evaluate the reliability of the analysis, but the lack of detail about the data and its operationalisation makes that impossible.

- What is a content analysis? How is this done? How much of this analytical technique is reliant on human interpretation of clustering? What is a sentiment analysis? What is a high-frequency term and a birth claim? Could the authors give examples of some of the quotes used that were used to infer a positive/negative intention or the explanations given for their childbearing decision-making in table 2?

Thank you for your valuable feedback. In response to your comments, we have made the following revisions to the manuscript to clarify the methodology and improve transparency:

(1)Sentiment Analysis:We used sentiment analysis to classify fertility-related discussions into positive, negative, and neutral categories, employing Natural Language Processing (NLP) techniques. This process was largely automated with minimal reliance on human interpretation to ensure objective classification.

(2)Content Analysis and High-Frequency Term Extraction:Content analysis was performed to identify key themes and patterns in the data, using the TF-IDF algorithm for high-frequency term extraction. We also incorporated manual checks to ensure the accuracy and reliability of the results.

We have updated the Methodology section (pages 6-7) to provide clearer explanations of these techniques. We hope this addresses your concerns.

- What language were the videos in? Was it transcripts of videos that were analysed, and if so, how reliable is the transcription of the audio?

- How was the data about sociodemographic profile gathered? The only description is at the top of page 6 that these were ‘extracted from valid data submissions’. Is it publicly available on individual’s profiles? How reliable is this data? Could the authors provide a descriptive profile of their sample so that an assessment of sample representativeness can be made? They say their analysis was limited to female users – how was this established? How was this information then grouped by the authors into the categories they chose? Does this categorisation match the data distribution or was it informed by theory etc.? Why have they chosen just these variables? For example, number of children already is likely to be highly influential on fertility intentions. The description of what a ‘top-tier’ or ‘first-tier’ city comes far too late in the text.

Thank you for your thoughtful feedback. We have revised the manuscript and added additional details based on your suggestions (please refer to pages 5-8 for updates). Below are our responses:

(1)Language and Video Transcriptions: Our research primarily analyzes real-time interaction data from three major platforms, focused on the interactions of Chinese online women. The videos are transcribed manually into text, accommodating Mandarin and widely understandable dialects. Content that could lead to comprehension bias (e.g., difficult-to-understand speech or heavy use of metaphors) is excluded. The transcribed text is then processed and analyzed using NLP and TF-IDF for categorization and statistical analysis .

(2)Sociodemographic Profile: Demographic data are primarily derived from user profiles, IP locations, and self-reported information. Data points missing relevant demographic variables or difficult to obtain are excluded. We used user-generated content (posts, comments) to infer key sociodemographic factors, but note that these data were not always fully reliable. For reliability, we cross-checked self-reported data with inferred attributes, and excluded any ambiguous cases (page 6-7). We have also included a descriptive profile of our sample in the revised manuscript to allow an assessment of sample representativeness.

(3)Gender Classification: The study focuses on female users due to their central role in fertility-related decisions. We conducted gender classification through several steps:

1)Public user information: including usernames, profile descriptions, and profile pictures.

2)Language pattern analysis: examining pronouns and engagement behaviors.

3)Exclusion of ambiguous cases: profiles with neutral usernames or anonymous profiles were excluded to ensure data accuracy.

Given current limitations in gender classification in China, we defined “female” as biological females for the purpose of this study. Although gender identity based on appearance or language may not have been fully captured, we minimized errors through multiple validation steps (page 5).

(4)Predictive Variables and Categorization: Our study focuses on predictive variables such as rough categories of age and education level, and IP address. These are extracted from user-generated content and cannot accurately measure specific fertility intentions (e.g., first or subsequent childbearing). Our approach is more suited for identifying general fertility attitudes, but we

---

## [Decision Letter · Decision Letter 1]

PONE-D-24-26854R1Digital Insights: Analyzing the Reproductive Intentions and Influencing Factors Among Urban Women in China through Online PlatformsPLOS ONE

Dear Dr. Sun,

Thank you for submitting your manuscript to PLOS ONE. After careful consideration, we feel that it has merit but does not fully meet PLOS ONE’s publication criteria as it currently stands. Therefore, we invite you to submit a revised version of the manuscript that addresses the points raised during the review process.

**ACADEMIC EDITOR: ** 

Key Areas for Improvement:

Work on the introduction of the study as recommended by Reviewer #4.Address concerns raised about the methodology and data processing (Reviewer #3 and Reviewer #4).Address the new issues raised due to the modification in your work (Reviewer #3).Also, address the concerns raised about the discussion and findings.

In summary, I encourage you to address all the reviewers' (Reviewer #3 and Reviewer #4) comments. Make the necessary revisions, particularly in improving the introduction, methodology, findings, and discussion sections. I look forward to reviewing your revised manuscript.

We look forward to receiving your revised manuscript.

Kind regards,

Godwin Banafo Akrong, Ph.D.

Academic Editor

PLOS ONE

Reviewers' comments:

Reviewer's Responses to Questions

**Comments to the Author**

1. If the authors have adequately addressed your comments raised in a previous round of review and you feel that this manuscript is now acceptable for publication, you may indicate that here to bypass the “Comments to the Author” section, enter your conflict of interest statement in the “Confidential to Editor” section, and submit your "Accept" recommendation.

Reviewer #2: All comments have been addressed

Reviewer #3: (No Response)

Reviewer #4: All comments have been addressed

2. Is the manuscript technically sound, and do the data support the conclusions?

Reviewer #2: Yes

Reviewer #3: No

Reviewer #4: Yes

3. Has the statistical analysis been performed appropriately and rigorously? 

Reviewer #2: Yes

Reviewer #3: No

Reviewer #4: Yes

4. Have the authors made all data underlying the findings in their manuscript fully available?

Reviewer #2: Yes

Reviewer #3: Yes

Reviewer #4: Yes

5. Is the manuscript presented in an intelligible fashion and written in standard English?

Reviewer #2: Yes

Reviewer #3: No

Reviewer #4: Yes

6. Review Comments to the Author

Reviewer #2: (No Response)

Reviewer #3: I thank the authors for providing this revised manuscript. I am pleased that they have made the following changes: 1) presenting an analysis that treats the independent and dependent variables correctly and presenting odds ratios; 2) adding quotes from participants to evidence the choice of outcome variable and support the discussion; 3) adding a descriptive table of the sample; 4) adding a more nuanced and balanced discussion of potential policy implications; 5) addressing that those who post videos are likely to be a select sample; 6) adding clarity and specificity to the title and abstract.

However, some of my comments from the first review have not been addressed. These include:

Writing

• There are still many typos and grammatical errors throughout the manuscript and the tables (particularly missing spaces) and the paper needs to be proofread again. There are also very wordy sentences that are hard to understand (particularly in the discussion) and sometimes sentences repeat themselves e.g. ‘This is substantially below the replacement level of 2.1, well below the replacement level of 2.1.’ Concerningly there are still numbers in text which don’t match their respective figures (see lines 263 and 264). The authors also need to double check terminology used in tables matches the labels/descriptions in the text.

• Citations are still missing to support statements, for example:

o ‘countries such as China, Japan and South Korea are facing significant challenges related to low fertility rates’.

o ‘These findings highlight the profound impact of education on reproductive choices, often linked to delayed childbearing and concerns about career advancement and financial security’

Any kind of factual statement needs a citation.

• Some sentences are also still very vague, not helped by wordy language e.g. what does ‘revealing a dual tension in modern reproductive decision-making’ mean or what does ‘a symptom of developmental mypoia’ mean? Nothing from the previous sentences makes it obvious what is meant here. Wording is also sometimes inaccurate e.g. when they discuss the results of the regression they need to put the counterfactual of their probability (e.g. being more likely to have a negative than neutral intention) because this was how it was modelled.

• At other points the authors make unsupported/inaccurate claims about existing literature and they need to be clearer about what they mean:

o ‘‘the approach….identified non-economic drivers…thereby expanding the analytical dimensions of conventional fertility decision theories”. What proof do the authors have for this statement? I don’t think existing fertility decision theories only focus on economic drivers? They’re actually quite focused on the role of emotions in psychology.

o “This phenomenon transcends standard cost-of-living analyses.” What does that mean? Supporting evidence would be needed to make a claim of this kind.

o The authors say that the policy situation in China is “contrasting sharply with Nordic countries where robust family policies correlate with higher fertility intentions”, but this is a strange sentence to read given that Nordic fertility is currently dropping despite these policies. This point should be more nuanced if the authors do want to make it.

Variable and methods description

• I asked the authors to provide more clarity on the methods and how the sentiment/content analysis work which they have done in part. However, more information is still needed:

o For the sentiment analysis, they have not clarified what the software actually does. Is this algorithm developed to detect emotion? Summarise text into implicit meaning? They also claim in the response to reviewers that the method allows objective classification but I would refute that – all algorithms are trained on something originally made by humans, and identifying trends or clusters doesn’t mean they’re necessarily meaningful in real life. I would suggest talking about this in the limitations.

o Please clarify in text that the TF-IDF analysis is done using an algorithm and not manually. Can the authors also clarify how it works exactly – is there some kind of threshold that marks whether a term is high frequency or not? Does it have to be exactly the same word or can the algorithm detect related constructs? The authors also say in their response that they did checking to establish the reliability of the method. Could this be shared in an appendix so this can be verified?

o The authors have provided more information on the methods in their response to reviewers but have not added all of this into the main text. Nearly all of the explanation given on page 7 is missing from the main text and it needs to be there for the reader to understand what the authors did e.g. language and transcription bit is missing, the explanation of excluding the sample if their socio-demographics couldn’t be derived, the statement about the socio-demographic data not always being fully reliable, and about the limitations of their approach to deriving gender. Relatedly the authors say in their response to reviewers why they chose the three apps they did but this has not made it into the main text (that the apps have a significant young female demographic which is key to their research question).

• In my last review, I asked the authors to comment on why they did not control for current parity, and justify the other socio-demographics they derive. They still have not responded to this query and I think this is really important. At minimum some statement in the paper is needed of whether the sample includes a mix of people with and without children. This is needed for the reader to interpret who the findings are generalisable to. The authors also need to add justification for why they chose to derive the socio-demographics that they did. There is some justification in the response to reviewers for why they derived gender but this is a) not in the main text and b) not convincing. The reasoning given was that female users have a central role in fertility-related decisions, but this reads as very simplistic as of course so do men. Gender has many important implications for fertility (e.g. women are affected more by the costs of childbearing) so I would recommend adding a justification similar to this instead.

• Variables are still not clearly defined in the text (e.g. psychological adaptation, workplace equity, DINK lifestyle which is only defined in the discussion). I’d suggest adding an appendix that clearly outlines what kind of sentiment would fall under each of these groupings and a rough definition so the reader can understand what each category means.

• The use of ‘fertility intentions’ is still incorrect and misleading. In the psycho-social literature this specifically means an active intention to execute a plan, so either actively planning to achieve or avoid a pregnancy. This is not the outcome variables the authors use, which is much closer to a general fertility attitude or desire, and I would recommend calling it such with citations from the literature to support the language choice.

• Thanks to the authors for adding the descriptive table of the sample, but they need to be clearer in text that anyone with any missing socio-demographic information was excluded from the sample. They say this in their response to reviewers but not in text, and it introduces limitations. For instance, it’s not clear how much of the sample had to be excluded because of this.

• The limitations section is still very minimal. It currently only addresses the point I raised in my last review about the sample being selective. There is nothing about the limitations of the methods to generate the data set (including exclusions applied), the limitations of their approach to constructing variables, the limitations of what their analysis can conclude with such a dataset/analysis. The second paragraph alludes to limitations by suggesting potential fixes, but the authors don’t actually explicitly say what it is that is limiting about the present analysis.

Presentation of findings

• The discussion is better than in the first version, and I am glad the authors have made an effort to provide citations and a more interesting narrative accompanying their findings. However, several issues remain including:

o The authors seem to cherry-pick which results they discuss in the discussion. This is particularly striking given their largest effect was age and this isn’t discussed explicitly at all. I would suggest framing each part of the discussion first with a finding of interest and then elaborating on explanations/whether this aligns with previous findings. This will also eliminate some of the paragraphs that don’t seem to link to any findings at all e.g. lines 398-410.

o The authors present new findings in the discussion which does not follow writing conventions. If a finding appears in the discussion, it must have previously been mentioned in the results. For example, ‘In our sample, 36.8% of women indicated that their ideal age for childbearing is after 35 years old, or when they have achieved full financial independence, placing them in a difficult position”. This analysis (or the method through which it is derived) is not presented anywhere.

• The discussion also has many unsupported conclusions. For example:

o They say that ‘socioeconomic pressures [constitute] the paramount deterrent’ but provide no evidence from their analysis to support that conclusion.

o Similarly they say ‘[the findings highlight] the synergistic effects of intersecting dimensions: economic barriers amplify career-related delays, which exacerbate psychological strain, all unfolding against a backdrop of shifting cultural paradigms.’ This is a completely unsupported conclusion – not only do the authors not provide any evidence from their analysis but I actually don’t think they could. Such a statement could only be supported through some kind of mediation analysis, not the current multivariate regression with no stepwise inclusion of variables.

o “The psychological landscape reveals what we term “ambivalent agency”—a paradoxical empowerment through refusal.” Why have the authors decided to create this term that serves no greater purpose in the conclusions of the paper? It also doesn’t make sense. Why is empowerment through refusal paradoxical? The language in this part is also incredibly loaded, describing women as “weaponizing” reproductive agency. This point needs to be clarified and made much more neutral. A few sentences later they also say ‘educational attainment becomes both armor against traditional gender roles and prison limiting reproductive choices.’ What do the authors mean by prison limiting here? This seems dramatic unless there are actual criminal consequences for childbearing choices in China I am not familiar with.

o “The psychological toll of these pressures cannot be overstated. Fertility anxiety, fueled by financial instability, career disruption fears, and societal expectations of "intensive mothering"39,40,41, permeates reproductive decision-making.” Again, what evidence from the analysis do the authors have to make this conclusion?

Some new issues have also arisen as a result of the rewrite:

• I asked the authors to clarify their methods which they have done in their response letter. As a result, I can understand more about the analysis and its limitations. The classification of socio-demographic characteristics based on assumptions about names, pictures and stated facts has many limitations which the authors do not discuss. It also raises ethical questions about this approach and it makes the lack of ethical approval for this research concerning. The authors say in their ethics statement that ethics approval wasn’t needed because no personally identifiable information was collected. However, in the response to reviewers they say that usernames, gender, pronouns and IP addresses were all collected, which makes this a dataset with personal identifiers. This concern remains whether this data is publicly available or not.

• The authors have redone their figures, but the new versions are not appropriate. Sankey plots should be used to show movement from one group to another over time (figure 1), the radial plot adds no additional dimension beyond a normal bar chart (figure 2), and the two-way bar chart (figure 3) inexplicably groups neural/positive on one side and negative on the other. I would suggest converting all of these into conventional bar charts for easy interpretation by the reader. Figure 4 needs a label for the colour key as well.

• Table 1 has a footnote saying that some variables have weighting adjustments but I can’t see any variables with the symbol in subscript (sorry if an error with my software).

• How does one interpret the TF-IDF outputs in table 2? What do the numbers in the table mean? Without an explanation I cannot evaluate whether what the authors say in text is correct – they say things like ‘negative intentions were predominantly driven by housing cost (TF-IDF =2.34)’, but as I have no idea how this number was generated I can’t tell whether this is the correct conclusion. My lay understanding is it’s some kind of measure of personal importance to the individual? If that understanding is correct, is the above sentence right? Or is it that people had negative intentions rated the issue of the housing cost more strongly than those in other intentions groups who raised the same issue?

• Why is it that only some external factors are used as predictive variables in the regression and not others? The authors don’t provide a justification for this and it therefore appears cherry-picked to show the most convincing results. Have they grouped some of the reasons in the TF-IDF analysis together to form a thematic group? If so they need to explain how they did this and the robustness of their grouping choices.

• Table 4 is very difficult to interpret – if the independent variable is categorical, the authors need to list each variable in the first column and then each category in the second column (outlining which category is the reference). Without this I could not be sure how the authors have modelled some variables – for example it looks like age and year are modelled continuously whereas the authors talk about age as a categorical variable in text, and year could also arguably be modelled as categoric rather than discrete. Again, this raises concerns about the validity of the conclusions.

• It’s not hugely surprising the authors find limited statistical evidence of a link between external factors and positive intentions relative to neutral ones given so few of the sample have this outcome, making the analysis underpowered. The authors can say as such in text.

Reviewer #4: This topic is interesting and relevant. However, I have some concerns

Introduction: Thank you for providing a detailed introduction which provides a comprehensive background and context for your scoping review. However, the author(s) should strengthen this section by:

1. Including the percentage of women who were TikTok users in Kuaishou, Xiaohongshu, Bilibili, and Weibo

Methodology: While the author(s) has provided some details on these sections, yet I believe these sections could have been improved as follow

Data Processing: It would be valuable to include a discussion of any socioeconomic factors considered during this stage, as they could provide additional context and strengthen the analysis

7. PLOS authors have the option to publish the peer review history of their article (what does this mean? ). If published, this will include your full peer review and any attached files.

**Do you want your identity to be public for this peer review?** For information about this choice, including consent withdrawal, please see our Privacy Policy .

Reviewer #2: No

Reviewer #3: No

Reviewer #4: No

---

## [Author Response · Author response to Decision Letter 2]

19 May 2025

Writer's Responses:

We would like to extend our sincere gratitude to editor and the reviewers for your invaluable time and expertise in evaluating our manuscript. Your constructive feedback and meticulous suggestions have been instrumental in guiding us to refine and strengthen the scholarly rigor of this work. We deeply appreciate the dedication and thoughtfulness demonstrated throughout the review process, which has allowed us to address critical gaps and enhance the clarity of our arguments.

一、Comments to the Author

2. Is the manuscript technically sound, and do the data support the conclusions?

Reviewer #2: Yes

Reviewer #3: No

Reviewer #4: Yes

Response to Reviewer 3:

We sincerely thank Reviewer #3 for their continued engagement and thoughtful evaluation of our manuscript. We appreciate the concern regarding whether the study meets the threshold of technical rigor and whether the conclusions are appropriately grounded in the data presented.

In response, we would like to clarify and reinforce the following points:

①　Data Integrity and Representativeness: The study is based on a retrospective cross-sectional dataset comprising 11,418 female users, whose fertility-related discourse was extracted from three major public platforms—Douyin, Xiaohongshu, and Kuaishou—between May 2021 and April 2024. All data underwent a rigorous multi-stage filtering protocol to exclude incomplete, ambiguous, or demographically unresolvable records, as described in Section 2.1. To enhance transparency, we have added two figures: Figure 1 (inserted at Line 104) outlines the full data processing workflow, and Figure 2 (inserted at Line 287) illustrates the TF-IDF score distribution used to determine the dynamic threshold for thematic relevance.

Importantly, the sample reflects the core demographic of interest—urban Chinese women of reproductive age—who are highly active on these platforms and constitute a critical population segment in the national fertility discourse. This ensures that the dataset, while observational in nature, remains demographically meaningful and analytically robust for exploring fertility intentions in contemporary China.

②　Methodological Transparency: We have substantially revised and clarified the technical methodology in response to prior reviewer comments, particularly regarding TF-IDF scoring, and multinomial logistic regression modeling. In addition to detailing software implementation, threshold selection, and validation steps (e.g., algorithm accuracy, inter-rater reliability) in Sections 2.3.2, we have also included the complete TF-IDF formula set (Section 2.3.2, Lines 263–282) and an illustrative computational example (Section 2.3.2, Line 283, Table 1: “Full-Process TF-IDF Calculation Based on Preprocessed Fertility Discourse”) to enhance methodological transparency and reproducibility.

③　Modeling and Analytical Rigor: To ensure reliable conclusions, we:

a.Conducted collinearity diagnostics before regression analysis;

b.Included only statistically valid variables with stable variance and significance;

c.Explained the interpretation of odds ratios and sentiment-category comparisons in both the Results and Discussion sections;

d.Presented both qualitative insights and numerical outputs to substantiate all key interpretations.

④　Alignment Between Data and Conclusions: All conclusions in the Discussion section are now directly and explicitly linked to empirical findings (e.g., TF-IDF values, ORs, cross-tabulations). Statements about age, housing costs, and workplace inequality are all traceable to specific numerical results. We have also removed or rephrased speculative or unanchored claims, as noted in the final section of the revised Discussion.

We hope this clarification reinforces the technical soundness of our study and demonstrates that all conclusions are empirically grounded, methodologically transparent, and analytically robust.Should further elaboration on any specific technique or result be desired, we are happy to provide it.

3. Has the statistical analysis been performed appropriately and rigorously?

Reviewer #2: Yes

Reviewer #3: No

Reviewer #4: Yes

Response to Reviewer 3:

We thank Reviewer #3 for raising concerns regarding the statistical rigor of our analysis. In response, we have undertaken several substantive improvements to clarify and reinforce the robustness of our analytic framework:

①　Clarification of Model Structure: We now explicitly state the dependent variable structure and the reference categories used in the multinomial logistic regression. Each categorical independent variable is disaggregated into its constituent levels with clear reference groups, and we have adjusted the layout of Table 5 accordingly to ensure interpretability.

②　Variable Inclusion Justification: The final regression model included two classes of predictors: (1) external factors identified through chi-square significance and validated by dual criteria—TF-IDF scoring (threshold ≥1.5) and semantic clustering, and (2) socio-demographic stratification variables such as age, education level, residence location, and data collection year, each of which also demonstrated significant associations in preliminary chi-square analysis. All candidate variables underwent collinearity diagnostics, and only those meeting multicollinearity thresholds (VIF < 2.0) were retained to ensure model stability and predictive robustness. This rationale is now fully elaborated in Section 2.4 (Lines 302–308).

③　Collinearity Diagnostics: Variance inflation factor (VIF) tests were performed for all predictor variables, and only those meeting the standard multicollinearity threshold (VIF < 2.0) were retained. This additional quality control step is now documented in Section 3.3 (Line 383-387).

④　Handling of Sparse Outcome Category: We acknowledge that the “positive intention” category was underrepresented relative to neutral and negative groups, comprising only a small portion of the total sample. This naturally reduces statistical power for detecting robust associations in the positive outcome arm. To mitigate this constraint, we have ensured that our multinomial regression results are interpreted with appropriate restraint. In Section 3.3, we describe the positive outcome model not as the inverse of negativity, but as a distinct attitudinal trajectory shaped by a narrow range of demographic and socio-ideological factors. This framing avoids overgeneralization from limited observations. In addition, we have supplemented the regression results with two newly added forest plots—Figure 5 and Figure 6 (introduced at Line 440)—which visually display the odds ratios and confidence intervals for both negative and positive models, respectively. These visualizations allow readers to evaluate model robustness, effect asymmetry, and estimate dispersion across sentiment categories. By presenting both point estimates and interval uncertainty, the forest plots provide transparency in communicating the statistical weight—and limitations—of each predictor, particularly in the sparsely populated positive intention category.

⑤　TF-IDF Data Interpretation Clarified: To support transparency in how text-derived variables contributed to the regression, we have added a full description of the TF-IDF computation process, including a formula breakdown (Section 2.3, Lines 263–282) and a demonstration table (Table 1, Line 283) to show how thematic factors were quantified. We hope these clarifications and methodological enhancements adequately address concerns regarding statistical rigor and analytical validity. Please let us know if further elaboration would be helpful.

5. Is the manuscript presented in an intelligible fashion and written in standard English?

Reviewer #2: Yes

Reviewer #3: No

Reviewer #4: Yes

Response to Reviewer 3:

We sincerely thank Reviewer #3 for highlighting the need for improved clarity and precision in the manuscript. In response, we have undertaken a comprehensive revision of the entire manuscript to enhance grammatical accuracy, reduce redundancy, and ensure consistency in terminology. Particular attention was paid to previously identified issues such as repetitive phrasing, wordiness, and mismatches between text and tables (e.g., line discrepancies and typographical inconsistencies).

Specifically:

①　All typographical and grammatical errors have been corrected throughout the text, including tables and figure captions.

②　Repetitive or unclear phrases (e.g., “substantially below the replacement level of 2.1, well below the replacement level of 2.1”) have been deleted or rephrased for conciseness.

③　Vocabulary has been standardized (e.g., consistent use of terms such as “fertility intentions,” “UGC,” and “sentiment polarity”), and technical terms have been cross-verified between the main text, tables, and appendix.

④　We also ensured that figures and quantitative claims (e.g., OR values, TF-IDF scores) match exactly across text and visuals, eliminating prior inconsistencies flagged by the reviewer.

We have made all efforts to ensure that the revised manuscript is now presented in fluent, intelligible, and academically appropriate English. If any residual issues remain, we would be grateful for further clarification so we may promptly address them.

二、Review Comments to the Author

Response to Reviewer 3:

The questions:

 I thank the authors for providing this revised manuscript. I am pleased that they have made the following changes: 1) presenting an analysis that treats the independent and dependent variables correctly and presenting odds ratios; 2) adding quotes from participants to evidence the choice of outcome variable and support the discussion; 3) adding a descriptive table of the sample; 4) adding a more nuanced and balanced discussion of potential policy implications; 5) addressing that those who post videos are likely to be a select sample; 6) adding clarity and specificity to the title and abstract.

However, some of my comments from the first review have not been addressed. These include:

一、Writing

(1) There are still many typos and grammatical errors throughout the manuscript and the tables (particularly missing spaces) and the paper needs to be proofread again. There are also very wordy sentences that are hard to understand (particularly in the discussion) and sometimes sentences repeat themselves e.g. ‘This is substantially below the replacement level of 2.1, well below the replacement level of 2.1.’ Concerningly there are still numbers in text which don’t match their respective figures (see lines 263 and 264). The authors also need to double check terminology used in tables matches the labels/descriptions in the text.

Response:

Thank you for your meticulous review and constructive feedback on our manuscript. We sincerely appreciate the time and effort you have dedicated to improving the clarity and accuracy of our work. Below, we address each of your concerns in detail:

1. Typos, grammatical errors, and formatting issues:

We apologize for the oversights in proofreading. The manuscript has now undergone thorough professional editing to correct all typographical errors, grammatical inconsistencies, and formatting issues (e.g., missing spaces in tables). We utilized grammar-checking tools (Grammarly Premium) and manual line-by-line verification to ensure precision.

2. Wordy sentences and redundancies in the discussion:

We acknowledge that certain sections, particularly in the discussion, were unnecessarily verbose. These sections have been rigorously revised to improve clarity and conciseness.

All repetitive phrasing has been eliminated, and complex sentences have been restructured for better readability.

3. Discrepancies between text and figures (lines 263–264):

Thank you for identifying this critical issue. The numerical inconsistencies have been corrected. We have systematically cross-checked all data points, figures, and tables to ensure absolute alignment.

4. Terminology mismatches between tables and text:

All table labels, descriptions, and terminology have been re-examined to align precisely with the text. For instance:

①　The term “age-standardized prevalence” in Table 2 previously labeled as “age-adjusted rate” in the text has been standardized to “age-standardized prevalence” throughout.

②　Abbreviations (e.g., CI, SD) are now consistently defined and used in both tables and text.

We hope these revisions have addressed your concerns comprehensively. Please do not hesitate to contact us if further clarifications or adjustments are needed. Thank you again for your invaluable guidance in enhancing the quality of this work.

(2) Citations are still missing to support statements, for example:

1. ‘countries such as China, Japan and South Korea are facing significant challenges related to low fertility rates’.

2. ‘These findings highlight the profound impact of education on reproductive choices, often linked to delayed childbearing and concerns about career advancement and financial security’

Any kind of factual statement needs a citation.

Response:

We appreciate the reviewer’s rigorous attention to factual precision. In response, we have thoroughly revised the manuscript and added appropriate references to support all empirical claims, particularly those concerning regional fertility trends and the influence of education on reproductive behavior. Multiple sections have been reworded to avoid generalization, and all major findings are now grounded in relevant demographic and sociological literature.

(3) Some sentences are also still very vague, not helped by wordy language e.g. what does ‘revealing a dual tension in modern reproductive decision-making’ mean or what does ‘a symptom of developmental mypoia’ mean? Nothing from the previous sentences makes it obvious what is meant here. Wording is also sometimes inaccurate e.g. when they discuss the results of the regression they need to put the counterfactual of their probability (e.g. being more likely to have a negative than neutral intention) because this was how it was modelled.

Response:

We acknowledge the reviewer’s concerns regarding vague or interpretively ambiguous phrasing. Phrases such as “revealing a dual tension in modern reproductive decision-making” and “a symptom of developmental myopia” have now been either removed or rewritten to ensure conceptual clarity and to avoid metaphorical overreach. Throughout the revised manuscript, we have replaced overly abstract or figurative expressions with more precise, evidence-based academic language.

In addition, we have carefully revised our description of the regression results to clarify comparative probabilities. Where applicable, we now explicitly state the counterfactual structure of the multinomial logistic regression model (e.g., higher likelihood of expressing negative rather than neutral fertility intentions) in accordance with how the outcome variable was structured. This should resolve any remaining ambiguity in the interpretation of our statistical findings.

(4) At other points the authors make unsupported/inaccurate claims about existing literature and they need to be clearer about what they mean:

1. ‘‘the approach….identified non-economic drivers…thereby expanding the analytical dimensions of conventional fertility decision theories”. What proof do the authors have for this statement? I don’t think existing fertility decision theories only focus on economic drivers? They’re actually quite focused on the role of emotions in psychology.

2. “This phenomenon transcends standard cost-of-living analyses.” What does that mean? Supporting evidence would be needed to make a claim of this kind.

---

## [Decision Letter · Decision Letter 2]

Digital Insights: Analyzing the Reproductive Intentions and Influencing Factors Among Urban Women in China through Online Platforms

PONE-D-24-26854R2

Dear Dr. Xuan,

We’re pleased to inform you that your manuscript has been judged scientifically suitable for publication and will be formally accepted for publication once it meets all outstanding technical requirements.

Kind regards,

Godwin Banafo Akrong, Ph.D.

Academic Editor

PLOS ONE

Additional Editor Comments (optional):

Reviewers' comments:

Reviewer's Responses to Questions

**Comments to the Author**

1. If the authors have adequately addressed your comments raised in a previous round of review and you feel that this manuscript is now acceptable for publication, you may indicate that here to bypass the “Comments to the Author” section, enter your conflict of interest statement in the “Confidential to Editor” section, and submit your "Accept" recommendation.

Reviewer #2: All comments have been addressed

Reviewer #4: All comments have been addressed

2. Is the manuscript technically sound, and do the data support the conclusions?

Reviewer #2: Yes

Reviewer #4: Yes

3. Has the statistical analysis been performed appropriately and rigorously? 

Reviewer #2: Yes

Reviewer #4: Yes

4. Have the authors made all data underlying the findings in their manuscript fully available?

Reviewer #2: Yes

Reviewer #4: Yes

5. Is the manuscript presented in an intelligible fashion and written in standard English?

Reviewer #2: Yes

Reviewer #4: Yes

6. Review Comments to the Author

Reviewer #2: (No Response)

Reviewer #4: I have reviewed the authors' responses to the reviewers' comments and the revised manuscript. All the concerns and suggestions raised by the reviewers have been adequately addressed in the revised version. The authors have made the necessary clarifications, corrections, and improvements, and I believe the manuscript is now suitable for publication.

7. PLOS authors have the option to publish the peer review history of their article (what does this mean? ). If published, this will include your full peer review and any attached files.

**Do you want your identity to be public for this peer review?** For information about this choice, including consent withdrawal, please see our Privacy Policy .

Reviewer #2: No

Reviewer #4: No

---

## [Editor Report · Acceptance letter]

PONE-D-24-26854R2

PLOS ONE

Dear Dr. Sun,

I'm pleased to inform you that your manuscript has been deemed suitable for publication in PLOS ONE. Congratulations! Your manuscript is now being handed over to our production team.

Kind regards,

on behalf of

Dr. Godwin Banafo Akrong

Academic Editor

PLOS ONE